# A Meta-Analysis of Social and Contextual Correlates of Migrant Adaptation to Living in Receiving Societies

Kinga Bierwiaczonek [1,2,3] ✉, Dinh H. Vu [3], Rongtian Tong [4], Mike W.-L. Cheung [5], Nora C. G. Benningstad [3], Evita van Duin [6], Karine Lindholm[3,7], Colleen Ward [8] & Jonas R. Kunst [9]

International migration has been consistently rising in modern times, and understanding what factors are associated with the successful inclusion of migrants is urgent. This meta-analysis helps pinpoint such factors by identifying the most robust social and contextual correlates of successful migrant adaptation to living in the receiving societies. Here, we meta-analyze 5,066 effects from 1,114 primary studies among 571,260 first-generation migrants, international students, business expatriates, and refugees. We show that migrant adaptation is most strongly negatively associated with the presence of stressors, especially acculturative stressors and perceived discrimination, and positively with the availability of social resources, especially feelings of connectedness with the social context and not feeling lonely. The role of variables related to culture learning, namely exposure to social groups within the new culture, and the distance between the new culture and one's heritage culture, was more limited. This pattern was found across the different migrant groups.

The number of international migrants, broadly defined as people living in a country different from their country of birth, has been consistently rising decade after decade in modern times. This trend has been especially pronounced over the past 20 years, with an increase from 173 million international migrants in 2000 to record levels of 270 million in 2020[1]. Scientists and stakeholders agree that to maximize the benefits of this global megatrend, understanding what psychosocial factors are associated with the successful and equitable inclusion of migrants is a matter of highest urgency[2,3]. Well-adapted migrants have better educational achievement[4,5], better work outcomes[6–9], higher capacity to behaviorally and cognitively fit into the receiving culture[10], and are overall more satisfied with their lives in the receiving country[11].

Given these social outcomes, the increase in worldwide migration has been accompanied by an exponential growth of studies on migrant acculturation and adaptation across various fields such as psychology, cultural studies and anthropology, organizational studies, communication studies, and sociology. The number of published studies on acculturation per year has been six times greater in the recent years (e.g., over 1200 studies in 2019) than in the beginning of the century (e.g., about 200 studies in 2001)[12]. While the sheer volume of research in this field might suggest that the factors crucial for migrants' adaptation are well understood, the body of literature is beset by inconsistent results and challenges in replication[13–16]. This situation blurs the state of knowledge and undermines its applied value for a topic of critical societal relevance. Here, we address this issue by presenting a comprehensive preregistered multi-level meta-analysis, analyzing data from 1114 primary studies and 5066 effects on migrant adaptation, covering 571,260 first-generation migrant participants arriving from 73 countries and residing in 64 countries.

Pertaining to the broader theoretical frameworks of acculturation[2,17,18] and intercultural contact[11], cross-cultural adaptation involves psychological and socio-cultural dimensions. Psychological

[1]Leibniz Institute for Psychology (ZPID), Trier, Germany. [2]Trier University, Trier, Germany. [3]University of Oslo, Oslo, Norway. [4]University of Washington, Seattle, WA, USA. [5]National University of Singapore, Singapore, Singapore. [6]Ghent University, Ghent, Belgium. [7]Norwegian Police University College, Oslo, Norway. [8]Victoria University of Wellington, Wellington, New Zealand. [9]BI Norwegian Business School, Oslo, Norway. ✉e-mail: kmb@leibniz-psychology.org

adaptation is acquired in a stress and coping process, involving dealing with the stressful elements of an intercultural transition. It is manifested in outcomes such as feelings of well-being and satisfaction when residing in the receiving culture. The psychological dimension of adaptation is broadly believed to be affected by two groups of factors: stressors related to migration (e.g., language barriers, discrimination experiences, decreases in social status post-migration) and resources that facilitate coping with these stressors (e.g., positive social interactions, friendships, diverse forms of social and organizational support).

By contrast, socio-cultural adaptation is acquired in a culture learning process, involving learning culture-specific norms, meanings, and behavioral skills. It manifests as being able to 'fit into' the new culture. The culture learning process behind socio-cultural adaptation is believed to be influenced by factors related to social learning (i.e., learning via exposure to the receiving society, such as through social interactions with receiving country nationals, but also co-nationals or individuals from other migrant groups) and learning generalization (i.e., being able to generalize culture-specific behaviors and knowledge acquired in one context to another cultural context, which depends, among others, on the degree of cultural distance, that is, dissimilarity between the receiving and heritage cultures)[11,19].

Cross-cultural adaptation theory proposes a long list of potential correlates of the two dimensions of adaptation, generally hypothesized to facilitate or hinder their underlying stress-and-coping and culture learning processes. While research on these correlates is extensive, empirical support for their relevance is inconsistent, and some of them may not be backed by solid evidence. Perhaps the most prominent example is the acculturation strategy of integration (i.e., an orientation toward both the migrant's heritage culture and the receiving culture). Long seen as a critical predictor of good adaptation[2,17,20], integration was recently revealed to have modest and highly heterogeneous associations with adaptation[14,15], triggering an ongoing debate on the practical relevance of this correlate[16-18,21]. This underscores the necessity of examining how robust the correlates put forward by the broader literature really are.

To complicate the picture even more, primary research on adaptation is scattered across several scientific fields with diverging theoretical assumptions and interests, often focusing on only one migrant population which, however, is rarely clearly delineated empirically (e.g., international students, migrants, expatriates, refugees)[13]. Despite offering focused and critical insights on specific factors relevant to adaptation, previous meta-analyses did not attempt to integrate this vast and dispersed literature due to their specialized foci. Specifically, one meta-analysis only covered correlates of one specific measure of adaptation (the Socio-Cultural Adaptation Scale)[19], two meta-analyses covered only one measure of adjustment for one specific migrant population (a scale of adjustment for expatriate managers)[7,8], and two other meta-analyses covered in great detail one variable for one migrant population (social support for international students[22] and for business expatriates[23]). These meta-analyses pointed to some important correlates of adaptation (e.g., perceptions of discrimination, $k = 7$, $r = -0.50$, 95% CI [−0.75, −0.25])[19]. However, their narrow scope did not allow for reliably establishing the relative importance of these correlates for migrants' outcomes across diverse groups and contexts.

Responding to the problem of mixed, scattered and fragmentary evidence in this field of research, our comprehensive meta-analysis is structured to aid in pinpointing the social and contextual factors robustly related to migrant adaptation. We address a broad question: Among the social and contextual correlates proposed by cross-cultural adaptation theory, which factors are the strongest and most robust positive correlates of migrant adaptation? Conversely, which ones are the strongest and most robust negative correlates across different migration contexts? Specifically, based on the frameworks reviewed so far, we focus on the relative importance of (a) social resources and stressors (both general and specific to migration, i.e., acculturative stressors), theorized to be related to stress-and-coping[11,13], and (b) exposure to social groups within the receiving culture and cultural distance between the culture of origin and the receiving culture, theorized to be related to culture learning[11,13,19]. By encompassing a broad spectrum of first-generation migrant populations, including migrants, business expatriates, international students, and refugees, situated across a variety of national, cultural, and political contexts, this approach subjects the identified correlates to a rigorous examination, aiming to identify those that demonstrate consistent patterns across diverse settings and groups.

## Results

The characteristics of the study pool are presented in Table 1. Figure 1 illustrates the geographical spread of the sample, and Figs. 1–8 summarize the main results. Whenever we refer to average effects obtained by subsetting the data, 95% confidence intervals around the mean effect are presented in the figures, indicating the precision of the estimate, and 90% prediction intervals, referring to the distribution of the population effect, are reported in the text. If both boundaries of a prediction interval are on the same side of zero, at least 95% of real-world effects can be expected to be so as well; if they are not, some real-world effects might go in opposing directions. Whenever we refer to comparisons between different categories of variables conducted as meta-regressions, only confidence intervals are reported because prediction intervals are not relevant for determining the significance of such comparisons. All tests of average effects $r$ and meta-regression coefficients $B$ are two-sided, whereas tests of heterogeneity $Q$ are one-sided as they test the null hypothesis that there is no significant heterogeneity. Importantly, our usage of the terms "effect" or "effect size" in the description of results is strictly due to meta-analytical conventions and is not intended to suggest causality (a point we return to in the discussion).

In the ensuing sections, we initially delineate the main categories of adaptation correlates considered in this study, namely exposure to different social groups within the receiving society (covering quantitative indicators of interaction with various social groups), cultural distance, perceived social resources (covering qualitative aspects of social interaction), and stressors. To maximize insights, we do so for the overall study pool and then for four different migrant groups: migrants, business expatriates, international students, and refugees. Subsequently, we delve into each category in detail to offer a nuanced understanding of these factors.

### Main variable categories

To estimate average effects across migrant groups, we first examined the overall patterns of associations between our key variable categories and both adaptation outcomes. In line with theory, cultural distance ($z(58) = -2.57$, $p = 0.006$, $r = -0.08$, 90% PI [−0.365, 0.202], $I^2 = 94.8\%$ for psychological adaptation; $z(185) = -9.68$, $p < .001$, $r = -0.16$, 90% PI [−0.454, 0.126], $I^2 = 87.79\%$ for socio-cultural adaptation) and stressors ($z(2,072) = -40.36$, $p < .001$, $r = -0.23$, 90% PI [−0.518, 0.049], $I^2 = 97.13\%$ for psychological adaptation; $z(456) = -20.55$, $p < .001$, $r = -0.26$, 90% PI [−0.592, 0.073], $I^2 = 94.19\%$ for socio-cultural adaptation) had significant negative overall associations with both psychological and socio-cultural adaptation. Exposure to social groups in the receiving society ($z(413) = 11.16$, $p < .001$, $r = 0.10$, 90% PI [−0.085, 0.292], $I^2 = 90.26\%$ for psychological adaptation; $z(245) = 8.10$, $p < 0.001$, $r = 0.14$, 90% PI [−0.156, 0.445], $I^2 = 87.13\%$ for socio-cultural adaptation) and perceived social resources ($z(1,031) = 29.43$, $p < 0.001$, $r = .24$, 90% PI [−0.044, 0.525], $I^2 = 93.77\%$ for psychological adaptation; $z(598) = 20.62$, $p < 0.001$, $r = 0.27$, 90% PI [−0.037, 0.574], $I^2 = 97.80\%$ for socio-cultural adaptation) had significant positive associations (Fig. 2). Notably, when considering the

**Table 1 | Characteristics of primary studies included in the meta-analysis**

| Studies on: | Studies with socio-cultural adaptation as the outcome | | | | Studies with psychological adaptation as the outcome | | | |
|---|---|---|---|---|---|---|---|---|
| | Cultural distance | Exposure | Social resources | Stressors | Cultural distance | Exposure | Social resources | Stressors |
| **All groups together** | | | | | | | | |
| Number of studies (k) | 98 | 79 | 180 | 210 | 39 | 144 | 401 | 719 |
| Number of participants (N) | 20 900 | 17 290 | 44 535 | 52 149 | 16 001 | 148 722 | 223 687 | 470 588 |
| % of male participants | 60.13 | 51.93 | 57.27 | 53.47 | 54.38 | 46.09 | 46.39 | 46.02 |
| Mean age of participants | 34.63 | 30.15 | 33.18 | 31.16 | 31.41 | 35.48 | 32.76 | 33.56 |
| Mean length of stay in the receiving country (months) | 37.65 | 39.67 | 48.66 | 49.34 | 30.66 | 108.97 | 82.02 | 95.19 |
| Number of countries of origin | 17 | 14 | 27 | 36 | 7 | 28 | 47 | 62 |
| Number of receiving countries | 21 | 22 | 31 | 34 | 13 | 30 | 44 | 53 |
| **Studies with migrants** | | | | | | | | |
| Number of studies (k) | 11 | 5 | 30 | 39 | 5 | 56 | 156 | 355 |
| Number of participants (N) | 3 521 | 1 367 | 6 636 | 9 489 | 7 593 | 114 180 | 150 358 | 333 875 |
| % of male participants | 57.32 | 50.12 | 52.63 | 48.64 | 49.27 | 42.48 | 40.29 | 41.04 |
| Mean age of participants | 33.58 | 46.05 | 36.12 | 36.91 | 34.17 | 44.96 | 38.71 | 38.55 |
| Mean length of stay in the receiving country (months) | 63.24 | 104.17 | 94.67 | 101.11 | 72.00 | 190.38 | 143.31 | 139.52 |
| **Studies with expatriates** | | | | | | | | |
| Number of studies (k) | 61 | 18 | 79 | 69 | 14 | 10 | 32 | 21 |
| Number of participants (N) | 12 252 | 2 622 | 16 335 | 13 950 | 1 946 | 1 622 | 5 595 | 4 898 |
| % of male participants | 64.98 | 66.48 | 67.30 | 64.53 | 59.76 | 66.23 | 54.95 | 60.09 |
| Mean age of participants | 39.42 | 42.22 | 39.09 | 37.81 | 39.94 | 41.54 | 40.54 | 38.67 |
| Mean length of stay in the receiving country (months) | 38.86 | 51.59 | 37.57 | 40.13 | 31.83 | 46.88 | 51.36 | 67.99 |
| **Studies with international students** | | | | | | | | |
| Number of studies (k) | 22 | 51 | 64 | 91 | 20 | 56 | 143 | 210 |
| Number of participants (N) | 4 603 | 12 437 | 20 548 | 26 001 | 6 462 | 17 211 | 42 206 | 69 498 |
| % of male participants | 48.96 | 47.70 | 48.85 | 48.65 | 51.89 | 45.57 | 47.43 | 46.64 |
| Mean age of participants | 22.81 | 24.15 | 24.03 | 24.28 | 24.49 | 24.07 | 24.20 | 24.49 |
| Mean length of stay in the receiving country (months) | 22.24 | 26.47 | 28.94 | 29.03 | 18.00 | 24.95 | 27.31 | 27.67 |
| **Studies with refugees** | | | | | | | | |
| Number of studies (k) | 1 | 3 | 4 | 6 | 0 | 15 | 65 | 121 |
| Number of participants (N) | 214 | 707 | 739 | 1 557 | 0 | 13 453 | 23 473 | 50 563 |
| % of male participants | 60.30 | 42.37 | 48.88 | 48.17 | N/A | 48.26 | 54.70 | 56.22 |
| Mean age of participants | 32.30 | 40.65 | 43.11 | 29.98 | N/A | 37.08 | 32.53 | 33.01 |
| Mean length of stay in the receiving country (months) | 25.60 | 39.81 | 84.65 | 33.23 | N/A | 37.72 | 47.34 | 62.02 |

The table displays the number of studies, participant demographics, and mean length of stay in primary studies grouped by migrant group (migrants, expatriates, international students, refugees), by the type of investigated outcomes (socio-cultural and psychological adaptation), and by the available correlates of adaptation (cultural distance, exposure, social resources, stressors). Please note that one study might be counted multiple times (e.g., if it reports both outcomes and/or multiple correlates).

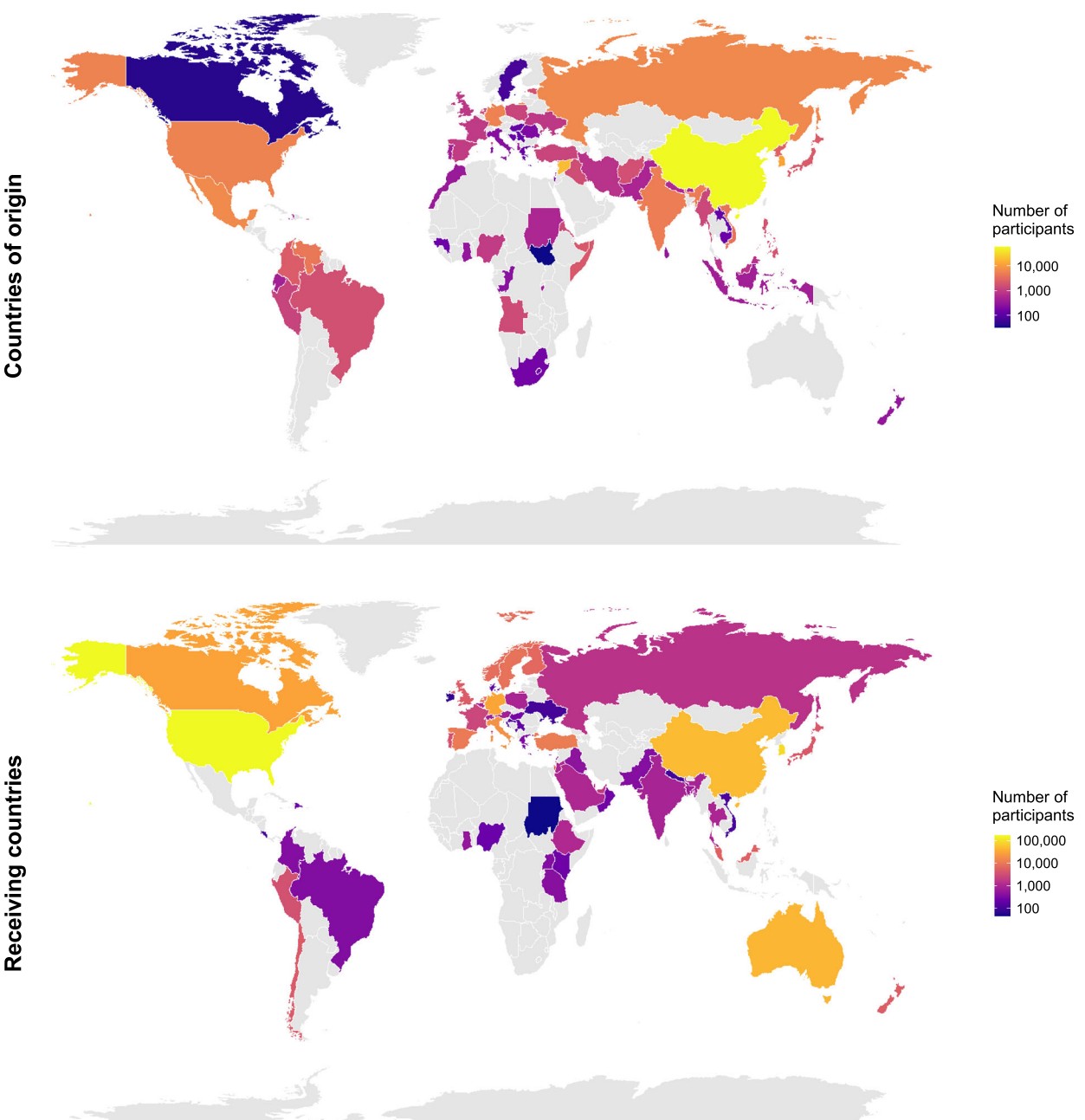

**Fig. 1 | Geographical coverage of the meta-analysis.** The number of participants per country of origin is presented in the top panel, and the number of participants per receiving country is presented in the bottom panel. Please note that this figure excludes samples with mixed (i.e., less than 60% of participants from one country) or unknown origins/receiving countries. This figure was created using the map_data function from the R package mapdata v.2.3.1[49].

absolute values of these effects, stress-and-coping factors (stressors, perceived social resources) had significantly larger overall correlations with both dimensions of adaptation than cultural distance and exposure to social groups, see Supplementary Table 1 for details.

Next, to estimate average effects in different migrant groups, we examined whether these general patterns held consistently across the four migrant populations (i.e., migrants, expatriates, international students, refugees). Indeed, the same pattern emerged when we analyzed the data separately for these groups (Fig. 3, see also Supplementary Table 2 for full results). Here, all effects were significant except for the correlation of cultural distance with psychological adaptation among migrants ($z(10) = -0.24$, $p = .814$, $r = -0.02$, 90% PI $[-0.322, 0.286]$, $I^2 = 98.41\%$) and expatriates ($z(14) = 0.15$, $p = 0.879$, $r = 0.01$, 90% PI $[-0.287, 0.303]$, $I^2 = 80.51\%$),

and the correlation of cultural distance with socio-cultural adaptation among migrants ($z(12) = -1.83$, $p = 0.067$, $r = -0.13$, 90% PI $[-0.528, 0.266]$, $I^2 = 80.51\%$). Note that among refugees, the correlation of cultural distance with socio-cultural adaptation (one primary effect) and psychological adaptation (no primary effects) could not be assessed due to lack of data. Stressors had significantly stronger negative effects for international students than for migrants in terms of psychological adaptation, $z(2068) = -3.57$, $p < 0.001$, $B = -0.05$, 95% CI $[-0.074, -0.022]$, with $R^2 = 1.32\%$ for the overall model, but there was no statistically significant difference for socio-cultural adaptation, $z(452) = -0.60$, $p = 0.550$, $B = -0.02$, with $R^2 = 0.67\%$ for the overall model. The remaining effects (perceived social resources, exposure to the receiving society, cultural distance) did showed no statistically significant differences between

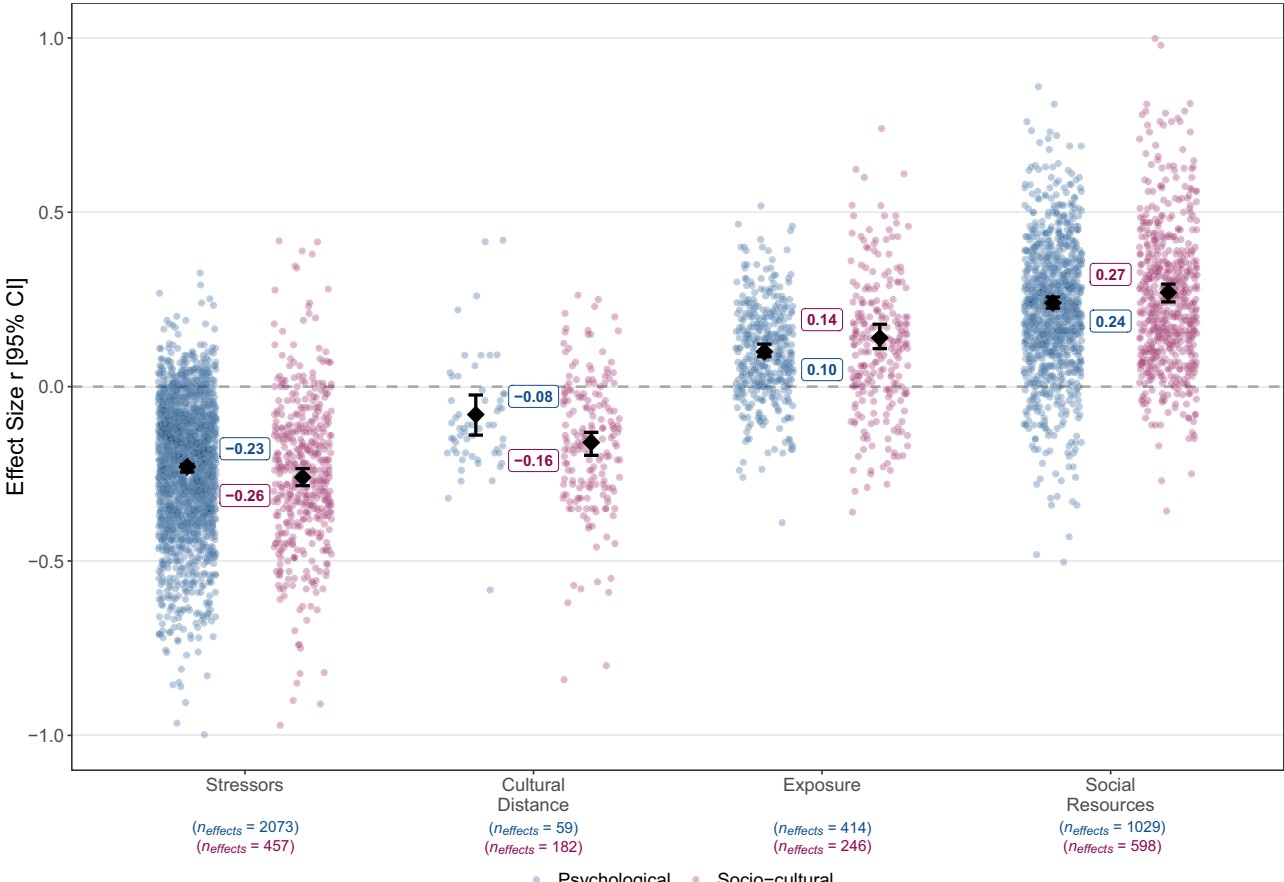

**Fig. 2 | Overall estimated true associations of the main categories of correlates (stressors, cultural distance, exposure to social groups, perceived social resources) with psychological (in blue) and socio-cultural (in pink) adaptation.** Dots refer to individual effect sizes, diamonds refer to average effects and whiskers indicate the 95% CIs. The included broad correlate groups are stressors, cultural

distance, exposure to social groups, and perceived social resources. This categorization explained a significant portion of effect variation, $Q_{explained}(3) = 132.53$, $p < .001$, $R^2 = 5.50\%$ for psychological adaptation, and $Q_{explained}(3) = 86.00$, $p < .001$, $R^2 = 6.19\%$ for socio-cultural adaptation. Full statistical details for each category are reported in the section in Supplementary Table 1.

these four groups for neither dimension of adaptation, see Supplementary Table 3 for full details.

## Heterogeneity

While these findings provide important insights into general patterns, our analyses also revealed substantial variation in effect sizes. In line with previous work[15,16], the meta-regression models with main categories of adaptation correlates revealed extremely high levels of unexplained true heterogeneity both for socio-cultural adaptation, $Q_{residual}(1484) = 116,960.04$, $p < 0.001$, $I^2 = 98.73\%$, and for psychological adaptation, $Q_{residual}(3574) = 677,310.5$, $p < 0.001$, $I^2 = 99.47\%$ (see also Supplementary Table 1). Moreover, all 90% prediction intervals within these categories included zero and thus pointed to inconsistent directions of true effects. Heterogeneity remained extreme when the main categories of correlates were tested within each migrant group in separate meta-regressions (see Supplementary Table 4), when we compared alternative migrant groups based on economic resources (Supplementary Tables 5,6), and when we regressed the effects on mean sample age and the percentage of male participants in the sample (Supplementary Table 7).

Against the backdrop of high heterogeneity, we moved to a more granular analysis examining specific components of each correlate group to identify nuanced patterns that might have been obscured in the higher-level data. Specifically, we analyzed each category of adaptation correlates separately (results reported in the next section), breaking them down into smaller subsets as outlined in the project

preregistration[24,25]. For the sake of conciseness, we present all the average effects in the figures, but in the main text we only describe the strongest and most consistent average correlations found across these subcategories. For details of all average effects and their distributions, please refer to the Supplementary Table 8. For the same reasons, we only provide full statistical information for significant comparisons between subsets, and only p-values for some non-significant comparisons. Full details of all meta-regression results can be found in Supplementary Table 9.

## Stressors

Among the different types of stressors (Fig. 4), the strongest negative correlation with psychological adaptation was found for acculturative stressors (an umbrella term referring to composite measures combining various stressors specific to migration into one score, used only in cases when scores for those specific stressors were not reported; see Supplementary Table 12), $z(333) = -28.15$, $p < 0.001$, $r = -0.35$, 90% PI [−0.630, −0.064], $I^2 = 95.81\%$. These were followed by occupational stressors, $z(67) = -9.02$, $p < .001$, $r = -0.28$, 90% PI [−0.591, 0.035], $I^2 = 93.35\%$, and COVID19-related stressors, $z(60) = -6.44$, $p < .001$, $r = -0.27$, 90% PI [−0.575, 0.035], $I^2 = 96.48\%$. For the latter two results, however, the 90% prediction interval included zero, suggesting that population effects can be inconsistent in their direction. Meta-regressions showed that the effect of acculturative stressors was significantly stronger than the effect of occupational stressors, $z(2063) = 2.34$, $p = 0.019$, $B = 0.05$, 95% CI [0.008, 0.095], but not

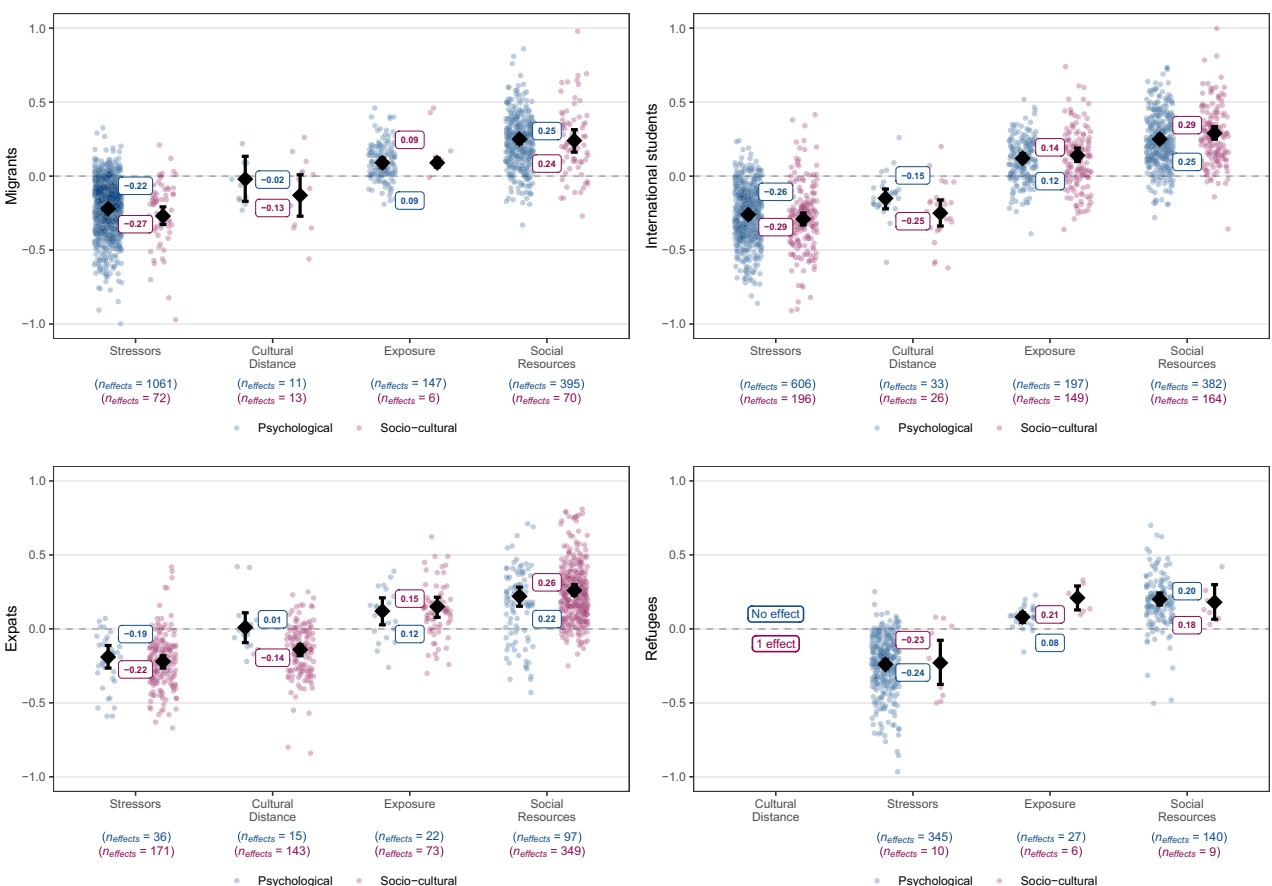

**Fig. 3 | Estimated true associations of main categories of correlates by migrant population with psychological (in blue) and socio-cultural (in pink) adaptation.** Dots refer to individual effect sizes, diamonds refer to average effects and whiskers indicate the 95% CIs. The included types of broad correlates are stressors, cultural distance, exposure to social groups, and perceived social resources. The distinction between migrant groups did not explain statistically significant portion of effect variation for the following categories: stressors, $Q_{explained}(4) = 6.34$, $p = .175$, $R^2 = 0.67\%$ for socio-cultural adaptation; exposure, $Q_{explained}(4) = 4.51$, $p = .341$, $R^2 = 0.00\%$ for psychological adaptation, $Q_{explained}(4) = 3.60$, $p = .463$,

$R^2 = 0.00\%$ for socio-cultural adaptation; social resources, $Q_{explained}(4) = 4.91$, $p = .293$, $R^2 = 0.07\%$ for psychological adaptation, $Q_{explained}(4) = 3.56$, $p = .453$, $R^2 = 0.00\%$ for socio-cultural adaptation. The distinction between migrant groups did not explain a statistically significant portion of effect variation for the following categories: stressors, $Q_{explained}(4) = 13.95$, $p = .007$, $R^2 = 1.32\%$ for psychological adaptation; cultural distance, $Q_{explained}(2) = 8.20$, $p = .017$, $R^2 = 17.94\%$ for psychological adaptation, $Q_{explained}(2) = 6.10$, $p = .047$, $R^2 = 1.71\%$ for socio-cultural adaptation. Full statistical details for each migrant group and correlate category are reported in Supplementary Table 2–6.

statistically significantly different from COVID-19-related stressors, $z(2063) = 1.74$, $p = 0.082$, $B = 0.05$, with $R^2 = 16.38\%$ for the overall meta-regression model.

For socio-cultural adaptation, the strongest negative correlates were perceived discrimination, $z(81) = -14.28$, $p < 0.001$, $r = -0.38$, 90% PI [−0.704, −0.063], $I^2 = 94.89\%$, general stressors, $z(19) = -6.03$, $p < 0.001$, $r = -0.30$, 90% PI [−0.590, −0.015], $I^2 = 88.95\%$, and acculturative stressors, $z(34) = -5.88$, $p < 0.001$, $r = -0.30$, 90% PI [−0.781, 0.188], $I^2 = 98.44\%$. Note, however, that for acculturative stressors the 90% prediction interval included zero, pointing to inconsistent true effects. In meta-regressions (Supplementary Table 9), the effect of perceived discrimination was statistically significantly stronger than the effect of general stressors, $z(450) = 2.88$, $p = 0.004$, $B = 0.14$, 95% CI [0.046, 0.240], and acculturative stressors, $z(450) = 2.38$, $p = 0.017$, $B = 0.09$, 95% CI [0.017, 0.171], with $R^2 = 10.16\%$ for the overall model.

**Perceived social resources**

Perceived social resources refer to participants' evaluations of the qualitative aspects of the resources available to them. First, we split this category by the type of resources (Fig. 5). Here, the largest average correlations were those of social connectedness (i.e., feeling

connected to one's social environment) with both psychological adaptation, $z(93) = 11.21$, $p < 0.001$, $r = 0.33$, 90% PI [0.046, 0.611], $I^2 = 91.15\%$, and socio-cultural adaptation, $z(11) = 7.08$, $p < 0.001$, $r = 0.38$, 90% PI [0.107, 0.652], $I^2 = 89.22\%$. The second strongest correlate was loneliness. This variable was reversed for easier comparison with other indicators of perceived social resources. Thus, the effect of *not* being lonely was $z(73) = 13.26$, $p < 0.001$, $r = 0.31$, 90% PI [0.043, 0.575], $I^2 = 93.94\%$ for psychological adaptation, and $z(10) = 3.60$, $p = 0.001$, $r = 0.30$, 90% PI [−0.168, 0.775], $I^2 = 97.35\%$ for socio-cultural adaptation, with the 90% prediction interval for the second test including zero and thus pointing to inconsistent true effects. The differences in effect size between connectedness and loneliness were not statistically significant for psychological adaptation, $z(1025) = -0.21$, $p = 0.836$, $B = -0.01$, with $R^2 = 3.23\%$ for the overall meta-regression model, and for socio-cultural adaptation, $z(592) = -0.28$, $p = 0.778$, $B = -0.02$, with $R^2 = 1.02\%$.

In the second step, we split perceived social resources by their source, that is, the social group that migrants received those resources from (Fig. 6). Whenever there was no specific group, like in the case of not being lonely, we classified the source as unspecific. The strongest correlations were observed when the resources came from supervisors (both in academic and work settings; $z(15) = 6.34$, $p < 0.001$, $r = 0.26$,

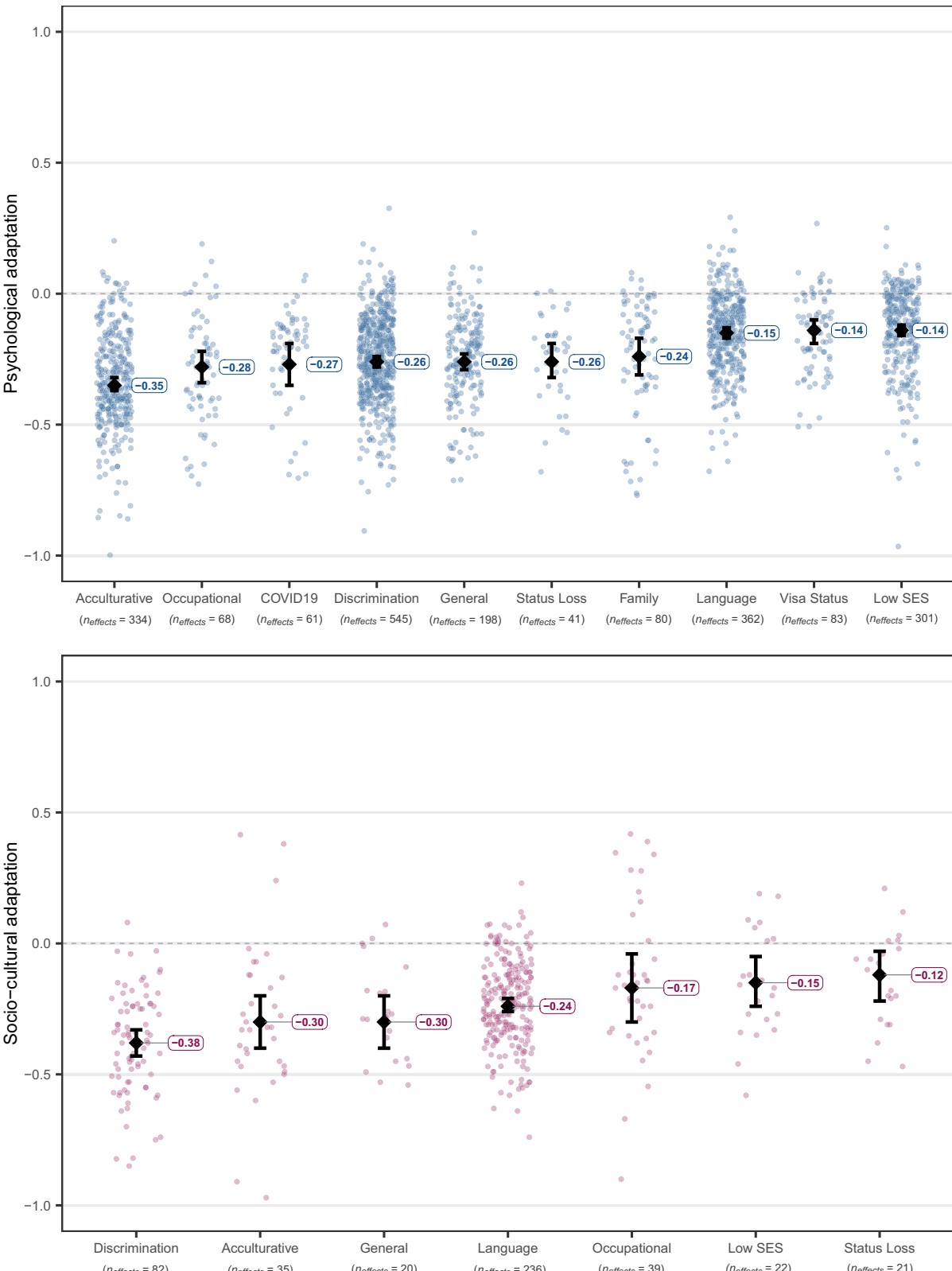

**Fig. 4 | Estimated true associations between different types of stressors and psychological (in blue) and socio-cultural (in pink) adaptation.** Dots refer to individual effect sizes, diamonds refer to average effects, and whiskers indicate the 95% CIs. The included types of stressor measures are: acculturative stressors, COVID-19 related stressors, occupational stressors, perceived discrimination, general stressors, family-related stressors, stressors related to loss of social status, undocumented visa status, low socio-economic status, and language barrier. This categorization explained a significant portion of effect variation, $Q_{explained}(9) = 306.61$, $p < 0.001$, $R^2 = 16.38\%$ for psychological adaptation, and $Q_{explained}(6) = 48.65$, $p < .001$, $R^2 = 10.16\%$ for socio-cultural adaptation. Full statistical details for each category are reported in Supplementary Tables 8–9.

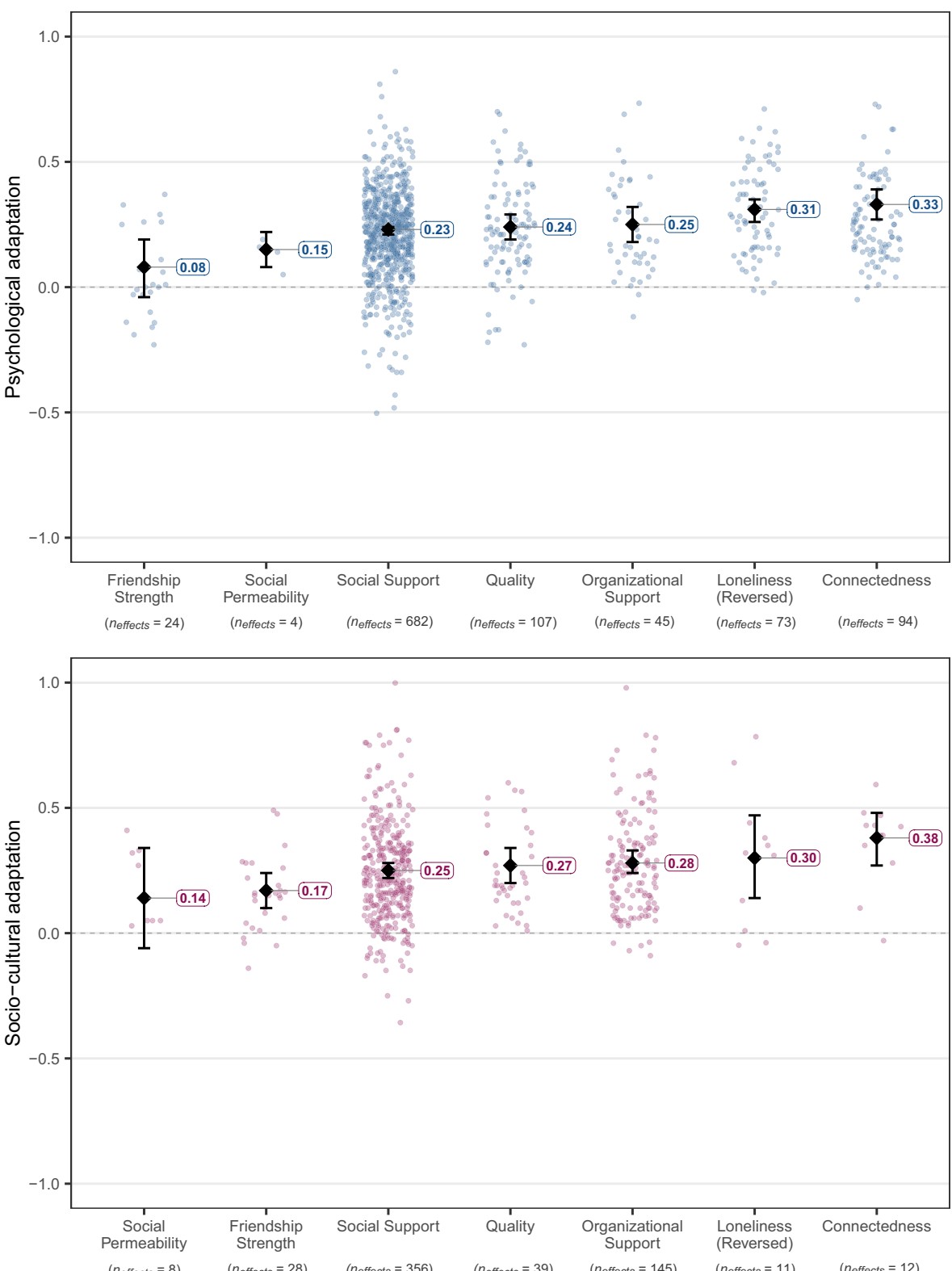

**Fig. 5 | Estimated true associations of different types of perceived social resources with psychological (in blue) and socio-cultural (in pink) adaptation.** Dots refer to individual effect sizes, diamonds refer to average effects, and whiskers indicate the 95% CIs. The included types of resource measures are: Self-reported strength of friendships, perceived permeability of the social environment, social support, organizational support, perceived quality of social relations, not being lonely, feelings of connectedness with the social environment. This categorization explained a significant portion of effect variation, $Q_{explained}(6) = 48.91$, $p < 0.001$, $R^2 = 3.23\%$ for psychological adaptation, and $Q_{explained}(8) = 17.71$, $p = .008$, $R^2 = 1.02\%$ for socio-cultural adaptation. Full statistical details for each category are reported in Supplementary Tables 8–9. For analyses differentiating between socio-emotional and instrumental support, see Supplementary Tables 10–11.

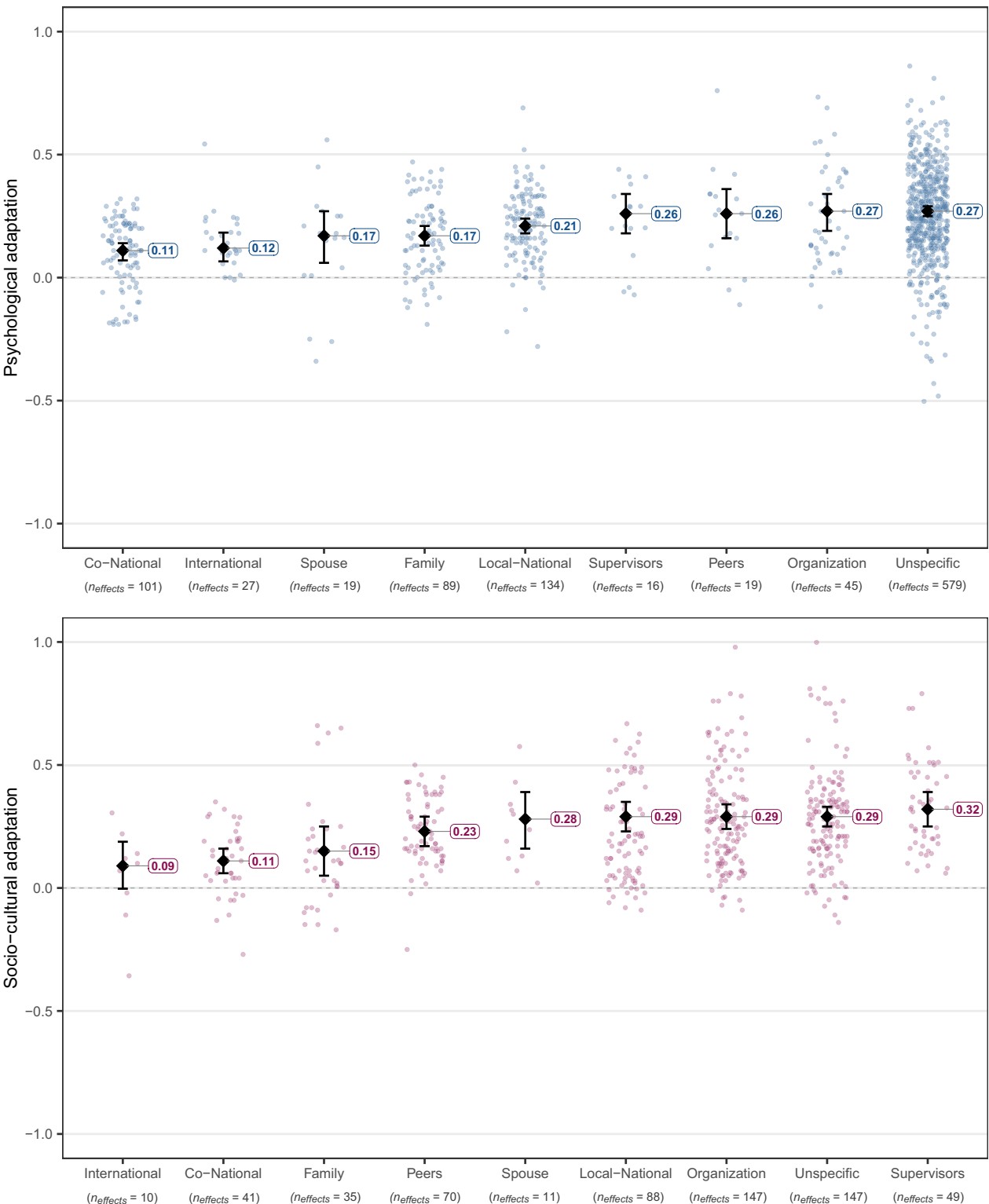

**Fig. 6 | Estimated true associations of perceived social resources from different sources (i.e., different social groups) with psychological (in blue) and socio-cultural (in pink) adaptation.** Dots refer to individual effect sizes, diamonds refer to average effects, and whiskers indicate 95% CIs. The included social groups social resources were assessed for are: Other internationals, co-nationals, receiving country nationals (labeled as local-nationals for conciseness), participants' spouses, family members in general, participants' supervisors, peers, organizations in general, and unspecified groups. This categorization explained a significant portion of effect variation, $Q_{explained}(8) = 67.09$, $p < 0.001$, $R^2 = 8.16\%$ for psychological adaptation, and $Q_{explained}(8) = 65.38$, $p < 0.001$, $R^2 = 7.42\%$ for socio-cultural adaptation. Full statistical details for each category are reported in Supplementary Tables 8–9.

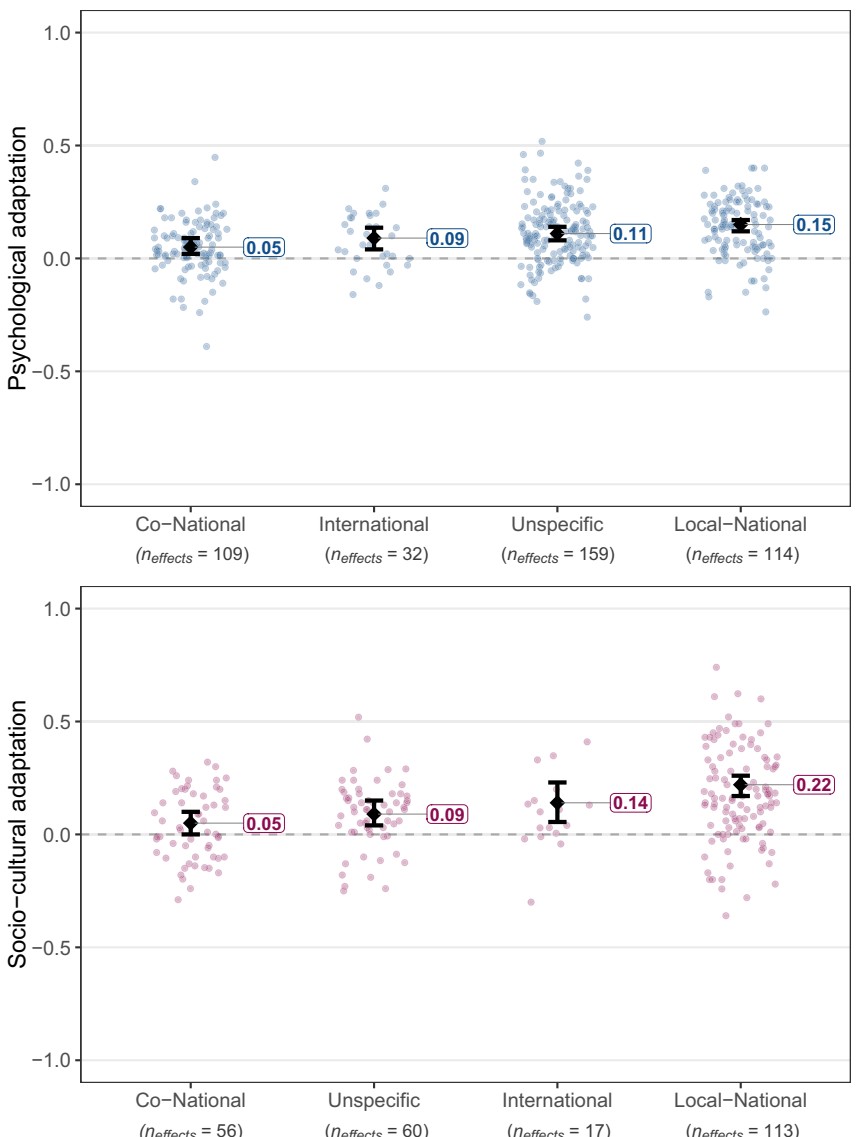

**Fig. 7 | Estimated true associations of exposure to different social groups with psychological (in blue) and socio-cultural (in pink) adaptation.** Dots refer to individual effect sizes, diamonds refer to average effects, and whiskers indicate 95% CIs. The included types of social groups exposure was assessed for are: Co-nationals, other internationals, receiving country nationals (labeled as local-nationals for conciseness), unspecified groups. This categorization explained a significant portion of effect variation, $Q_{explained}(3) = 38.82$, $p < .001$, $R^2 = 7.03\%$ for psychological adaptation, and $Q_{explained}(3) = 44.52$, $p < .001$, $R^2 = 9.04\%$ for socio-cultural adaptation. Full statistical details for each category are reported in Supplementary Tables 8–9.

90% PI [0.040, 0.484], $I^2 = 79.18\%$ for psychological adaptation; $z(48) = 8.94$, $p < 0.001$, $r = 0.32$, 90% PI [0.037, 0.605], $I^2 = 91.57\%$ for socio-cultural adaptation), from organizations (mainly employer or university, in five cases other organizations such as NGOs; $z(44) = 7.14$, $p < 0.001$, $r = 0.27$, 90% PI [−0.064, 0.596], $I^2 = 89.63\%$ for psychological adaptation; $z(146) = 11.70$, $p < 0.001$, $r = 0.29$, 90% PI [−0.039, 0.617], $I^2 = 96.14\%$ for socio-cultural adaptation) and from unspecific sources ($z(581) = 25.78$, $p < 0.001$, $r = 0.27$, 90% PI [−0.027, 0.558], $I^2 = 93.23\%$ for psychological adaptation; $z(147) = 13.77$, $p < 0.001$, $r = 0.29$, 90% PI [−0.023, 0.605], $I^2 = 96.87\%$ for socio-cultural adaptation). However, in the case of resources coming from organizations and from unspecific sources, 90% prediction intervals included zero, suggesting an inconsistent direction of true effects. The effects of these two sources did not show statistically significant differences in size from the effects of resources coming from supervisors neither for psychological nor socio-cultural adaptation, see Supplementary Table 9 for details.

## Exposure to social groups

Differently than social resources, which refer to participants' evaluations of the qualitative aspects of resources available to them from different social groups (e.g., quality of contact or support), exposure to social groups refers to quantitative indicators (e.g., frequency of contact, ethnic density; Fig. 7). Here, the strongest correlation was between the quantity of interaction with nationals of the receiving country and socio-cultural adaptation, $z(112) = 8.55$, $p < 0.001$, $r = 0.22$, 90% PI [−0.112, 0.542], $I^2 = 89.75\%$, with prediction intervals including zero and thus pointing to inconsistent effects. This effect was significantly larger than the effects of the quantity of interaction with co-nationals (i.e., individuals from the same country participants migrated from), $z(242) = −242$, $p < 0.001$, $B = −0.18$, 95% CI [−0.228, −0.123], and unspecific groups, $z(242) = −3.37$, $p < 0.001$, $B = −0.10$, 95% CI [−0.160, −0.042], but showed no statistically significant difference from interaction with other internationals, $z(242) = −1.53$, $p = 0.126$, $B = −0.07$, with $R^2 = 9.04\%$ for the meta-regression model. For psychological

adaptation, quantity of interaction with nationals of the receiving country, $z(113) = 10.03$, $p < 0.001$, $r = 0.15$, 90% PI [−0.019, 0.310], $I^2 = 81.08\%$, had a significantly larger effect than with co-nationals, $z(410) = −6.09$, $p < 0.001$, $B = −0.10$, 95% CI [−0.132, −0.068], and with other internationals, $z(410) = −2.17$, $p = 0.030$, $B = −0.05$, 95% CI [−0.104, −0.005], but showed no statistically significant difference from interaction with unspecific groups, $z(410) = −1.63$, $p = 0.103$, $B = −0.03$, with $R^2 = 7.03\%$ for the meta-regression model.

### Cultural distance

We compared the effects of self-rated cultural distance with the effects of objectively measured cultural distance whenever available in the reviewed studies. Self-rated distance referred to perceived differences between the heritage culture and the receiving culture as reported by participants themselves. Objectively measured distance referred to indexes tapping into cultural differences between two given countries based on external criteria, such as differences in cultural values from external databases (e.g., Kogut & Singh's index[26]). In all cases, the specific indexes were selected and applied by the authors of the primary studies.

In the case of psychological adaptation, the effect of objectively measured distance was statistically non-significant, $z(8) = −0.53$, $p = 0.595$, $r = −0.02$, 90% PI [−0.189, 0.149], $I^2 = 93.07\%$, whereas the effect of self-rated distance was negative and significant, $z(49) = −2.61$, $p = 0.009$, $r = −0.09$, 90% PI [−0.395, 0.215], $I^2 = 93.59\%$. However, both effects did not show statistically significant difference in size, $z(57) = −0.71$, $p = 0.476$, $B = −0.05$, 95% CI [−0.206, 0.096], $R^2 = 0.00\%$. For socio-cultural adaptation, self-rated distance also had a significant negative effect, $z(142) = −10.36$, $p < 0.001$, $r = −0.20$, 90% PI [−0.475, 0.084], $I^2 = 87.49\%$, with the prediction interval pointing to inconsistent true effects. This effect was significantly stronger, $z(184) = 3.30$, $p < 0.001$, $B = 0.13$, 95% CI [0.052, 0.204], $R^2 = 5.71\%$, than the effect of objectively measured distance, $z(42) = −2.00$, $p = .045$, $r = −0.08$, 90% PI [−0.401, 0.250], $I^2 = 87.92\%$.

## Discussion

Across 5066 effects from 1,114 primary studies among 571,260 first-generation migrants, international students, business expatriates, and refugees, the degree of cross-cultural adaptation was most strongly associated with factors related to stress-and-coping. Specifically, the presence of stressors, especially acculturative stressors and perceived discrimination, was negatively associated with adaptation, whereas the availability of social resources to deal with stressors, especially feelings of connectedness with the social context and not feeling lonely, were positively associated with it. Although these factors have often been assumed more relevant to psychological adaptation, theorized to be the direct outcome of the stress-and-coping process[11], in practice they turned out to be comparably relevant to socio-cultural adaptation, that is, behavioral and cognitive fit into the receiving culture. The role of variables theorized to be related to culture learning, namely exposure to social groups that can model or explain the new culture, and the cultural distance between the new culture and one's heritage culture, turned out to be more limited yet still having effects for both dimensions of adaptation. Of note, and in line with theoretical predictions, these effects were slightly larger for socio-cultural adaptation, especially in the case of self-rated cultural distance and exposure to local nationals.

Interestingly, this pattern was repeated in all four migrant groups commonly studied in the intercultural field. This is an important insight, since many have theorized that contexts and experiences of migration change drastically depending on whether one moves countries as an economic migrant, a refugee, an international student, or a business expatriate, and that the factors determining the successful adaptation also differ between these groups[11,13,27]. Our analysis, broadly speaking, finds little evidence for this notion. Of course, the exact patterns may show smaller differences between groups, for instance stressors seem to affect international students stronger than

migrants. Other differences might emerge when testing group differences for narrower subsets of adaptation correlates, such as different sources of support, which is beyond the scope of this paper. It is also crucial to note that some groups may be more exposed to specific factors than others due to varying contexts of reception (e.g., greater stress experienced by refugees owing to unwelcoming policies). Finally, for the sake of comparability, this meta-analysis did not include factors specific to some groups that might be of importance (e.g., exposure to war for refugees). Notwithstanding these qualifications, the key message remains the same regardless of who migrates: Experiencing stressors, especially acculturative stressors and discrimination, is associated with greater migration difficulties, while having access to social resources, especially feeling connected and not lonely, is associated with more positive migration experiences.

Crucially, heterogeneity was extreme in all analyses, which was expected considering similarly high levels of heterogeneity in existing meta-analyses on acculturation[15,16] and the diversity of cultural contexts of origin and reception covered by this study. Thus, based on prediction intervals, we identified those correlates of adaptation that showed consistent effects despite the fact that the primary studies behind them represented a myriad of different migration contexts, populations, and measures. If 90% prediction intervals exclude zero, they indicate that the real-world associations can be expected to go in one direction, either positive or negative, at least 90% of the time and regardless of contexts, which is critical for practitioners and policymakers. Of the 99 main meta-analytic effects tested here, only 16 fulfilled this condition (see Supplementary Table 1). Of these, seven effects exceeded the absolute value of 0.30, a threshold recently found to correspond with a large correlation in psychological research[28–30]. For psychological adaptation, these were social connectedness, not being lonely, and the presence of acculturative stressors. For socio-cultural adaptation, these were perceived discrimination, social connectedness (the two factors showing the strongest effect in this meta-analysis), general stressors, and social resources obtained from work and study supervisors.

These seven factors appear to be the strongest and most consistent correlates of cross-cultural adaptation. It must be acknowledged, however, that some of them might be so because of their conceptual and semantic proximity to adaptation measures. For instance, in light of previous research and theorizing[31,32] it is plausible that being connected to one's social environment and having enough people around not to feel lonely is crucial to good adaptation. Yet, this observed association might be exacerbated by the fact that measures of connectedness, loneliness, and psychological and socio-cultural adaptation are likely to tap into related feelings. Similarly, while encountering a lot of acculturative stressors at once, as conveyed by composite acculturative stress scales[33,34], is likely to translate into high levels of stress and thus poor long-term psychological adaptation[11,35,36], this effect might be inflated by the fact that measures of both acculturative stressors and psychological adaptation often include conceptually similar concepts[37] such as homesickness[38,39] assessed with semantically similar measures. We have discussed the implications of this overlap elsewhere[40], and we summarize the main conclusions in Supplementary Note 1.

One factor that does not seem to bear such limitations is perceived discrimination. Whereas the negative effects of perceived discrimination are in line with previous research[19,41], our meta-analytical test is unique in that it directly shows how robust these effects are with large-scale data, especially for socio-cultural adaptation. Here, however, a different limitation of the acculturation literature must be acknowledged, namely a critical lack of experimental studies[15], meaning that this meta-analysis is almost entirely based on observational studies and correlations (with the exception of four intervention studies)[40]. These provide no information about the causal links between the variables discussed here, including perceived discrimination. In other words, migrants who are discriminated against

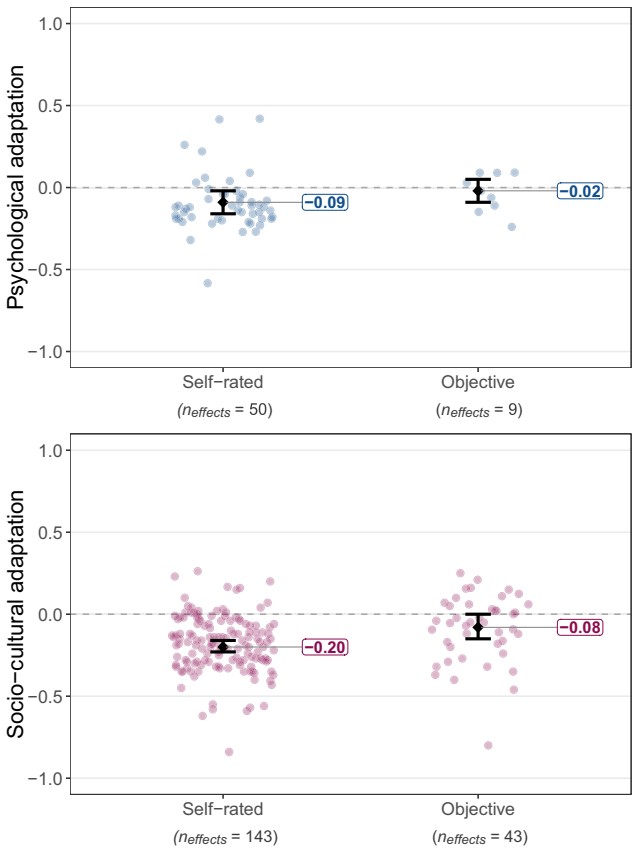

**Fig. 8 | Estimated true associations of self-rated and objectively measured cultural distance with psychological (in blue) and socio-cultural (in pink) adaptation.** Dots refer to individual effect sizes, diamonds refer to average effects, and whiskers indicate 95% CIs. The included types of cultural distance measures are self-rated distance (as reported by the participants themselves) and objectively measured cultural distance (calculated by the authors of the primary studies based on external criteria; labeled as objective for conciseness). This categorization did not explain a statistically significant portion of effect variation for psychological adaptation, $Q_{explained}(1) = 0.51$, $p = 0.476$, $R^2 = 0.00\%$, but it explained a statistically significant portion of effect variation for socio-cultural adaptation, $Q_{explained}(1) = 10.90$, $p < .001$, $R^2 = 5.71\%$. Full statistical details for each category are reported in Supplementary Tables 8–9.

focus on, as most instruments used to assess supervisory support combined socio-emotional and instrumental elements or lacked enough detail for clear classification (see Supplementary Table 12). Future studies that isolate these specific forms of support and test how each predicts migrant adaptation would greatly advance this under-researched area.

When interpreting the results of the present meta-analysis, it is important to acknowledge that despite the diverse range of origin and reception contexts covered, the Global North is overrepresented in our data (see Fig. 1). This geographical imbalance limits the generalizability of our findings, for instance regarding migration trajectories within the Global South. Future research should address this limitation by focusing more on the underrepresented regions and directly examining the role of country-level factors through comprehensive multi-level analyses, which would provide more nuanced insights into potential contextual variations in acculturation processes. Moreover, our analysis focused on the most influential framework within acculturation research, which encompasses psychological and sociocultural dimensions of adaptation. However, physical health remains a significantly understudied and undertheorized dimension of acculturation. Future theories, primary work, and meta-analyses could benefit from including physical health as an additional outcome.

Despite these limitations, our meta-analysis has provided a comprehensive, large-scale overview of factors associated with migrant adaptation and identified the largest and most consistent effects: perceived discrimination, acculturative stressors, not being lonely, social connectedness, and support from supervisors. This is an important insight for policy and practice, as these factors show considerably stronger and more consistent associations with adaptation than other correlates put forward by the acculturation theory, including those traditionally considered critical to migrant outcomes (i.e., migrants' integration orientation, for which average correlations oscillate around .10)[15]. If future longitudinal and experimental research supports the causality of these effects, they can inform actionable recommendations and programs to support thousands of migrants around the world in their efforts to build themselves happy, successful, and well-adapted lives within their receiving societies.

## Methods

### Transparency of methods and materials

This study is part of a larger pre-registered meta-analytical project consisting of cross-sectional, longitudinal, and contextual analyses of correlates of cross-cultural adaptation of first-generation migrants. The project was preregistered at https://osf.io/qc9h2/files/nb3rs on January 31, 2023, and updated on November 19, 2024[25]. The analyses presented here are described under Work Package 2, which corresponds with the cross-sectional component of the project, whereas the longitudinal and contextual components were preregistered as Work Packages 3 and 4 and will be reported separately. Registered deviations from the original plan relevant to Work Package 2 consisted of: dropping family variables other than family support from this study, as they did not align with the theoretical distinction between stressors, resources, exposure and cultural distance; dropping analyses and data items referring to mainstream society members due to a lack of data allowing to compare them to migrants; dropping social cognition as a category of adaptation correlates due to lack of data; improving quality by adding two additional exclusion criteria for primary studies (irrelevant intervention studies, studies based on the same data and reporting the same variables as other included studies); adding a moderator distinguishing between socio-emotional and instrumental support; and specifying the risk of bias assessment tool which was missing. All analyses referring to added data items are reported in the current paper and/or Supplementary Tables. Analyses referring to dropped data items are not reported due to a lack of data.

might be given less chances to participate in the receiving society and thereby improve their own socio-cultural adaptation, but it is also possible that migrants who do not fit in with the receiving society end up being discriminated as a consequence[19]. While we did not find any experimental studies with first-generation migrants that would clarify the causal direction, future meta-analyses of longitudinal effects might provide valuable insights.

Perhaps the most surprising of the seven strongest and most consistent results of this meta-analysis is the link between adaptation and social resources received specifically from participants' supervisors, both in work- and study settings. This finding points to a particular value of feeling supported by people in authority and, arguably, those having the means to facilitate migrants' adaptation experience. To the best of our knowledge, this effect has not been identified in any previous meta-analysis, including those that took an organizational angle[7,8]. Assuming that this effect reflects a causal relation – which is yet to be demonstrated – it might be easily actionable for intervention purposes: If support from a supervisor makes a considerable difference to migrant adaptation, then training supervisors to provide it in the most efficient way might go a long way. Unfortunately, it remains unclear what kind of support supervisors would be best advised to

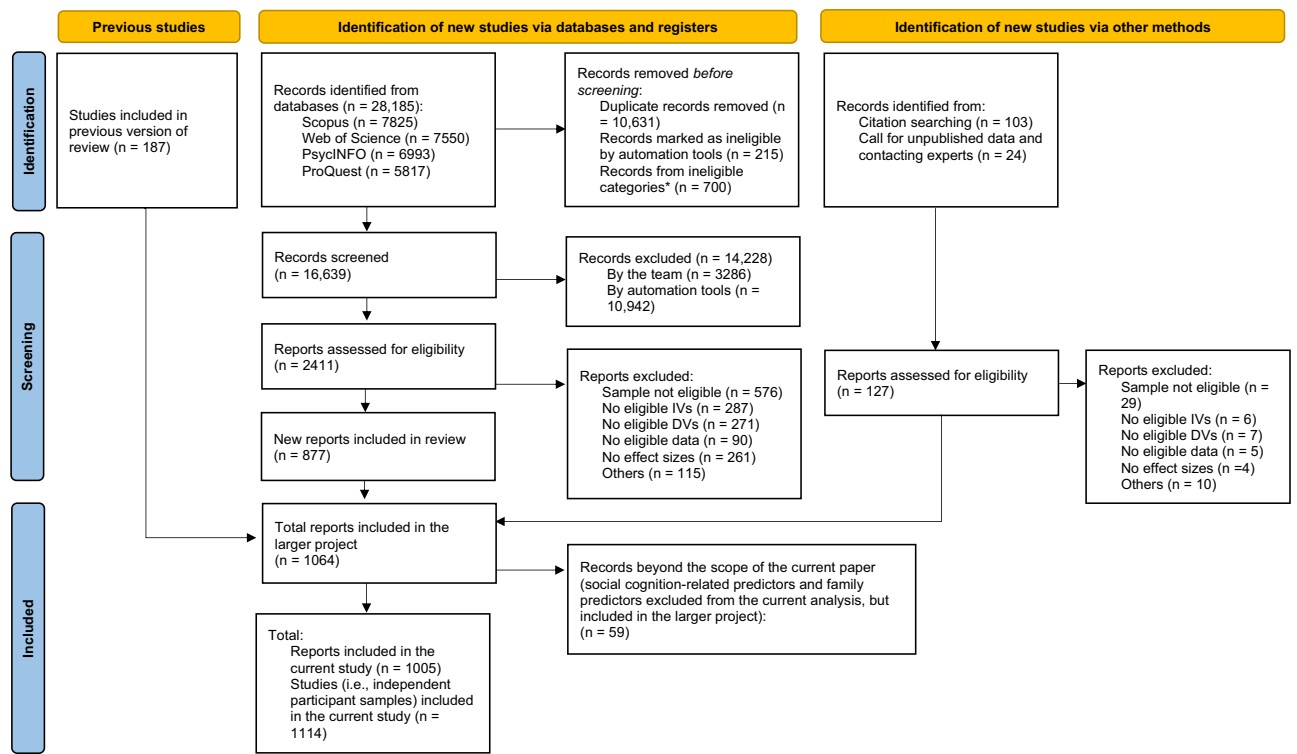

**Fig. 9 | PRISMA 2020 flow diagram summarizing literature search.** All sources were consulted on June 20, 2022, except for ProQuest, consulted on June 24, 2022. Call for unpublished studies was sent on May 9, 2023. This figure is based on the PRISMA[43] template (https://www.prisma-statement.org/prisma-2020-flow-diagram) licensed under CC BY 4.0 (https://creativecommons.org/licenses/by/4.0/).

No statistical method was used to predetermine sample size. No data were excluded from the analyses. Due to reference number limitations, bibliographic references of the included reports are available in Supplementary References. Below, we provide general information about our methods, and technical details can be found in Supplementary Methods. Additionally, methodological issues found in the meta-analyzed literature, including field-specific practices and conceptual overlap between variables, are summarized in Supplementary Note 1.

### Eligibility criteria
To be eligible for this study, a primary study needed to fulfill the following criteria: (1) use a quantitative design; (2) be conducted no earlier than 1988 (Black's study is considered the first study with a theoretical framing fitting this meta-analysis)[42]; (3) be available in English; (4) include an eligible sample consisting of participants in international mobility; (5) include at least one eligible measure of cross-cultural adaptation; (6) include at least one eligible measure of correlates of adaptation (for a list and sample items, see Supplementary Table 12), (7) report at least one correlation between a measure of adaptation and a measure of adaptation antecedent or other statistical information sufficient to estimate at least one such correlation.

### Literature search
Literature search followed the PRISMA guidelines[43] and is summarized in Fig. 9 (see also Supplementary Table 17 for the PRISMA checklist). Studies dating from the period 1988–2014 were included from an existing database by Bierwiaczonek[13,44]. Studies dating from 2014 or later were identified using a new literature search.

### Data screening
All hits from the databases covered by the core literature search ($k = 16{,}639$) were screened using ASReview Lab program[45]. Three members of the project team (one holding a PhD in psychology, two

holding master's degrees) took turns reviewing titles and abstracts. All the abstracts marked as irrelevant were then checked independently by the fourth team member (holding a master's degree). All discrepancies were discussed and resolved by the team.

### Coding
Full texts of the retained records were screened and coded by nine trained members of the project team (either holding a master's degree or master students). Each full text was screened and coded independently by two team members, resulting in a 77.29% agreement. Full texts of any studies that generated discrepancies between coders were re-analyzed in weekly project team meetings. Any discrepancies or difficulties in the coding process were then discussed and resolved by the first author and the project team. Additionally, any outliers were checked by three members of the coder team prior to running the analyses.

### Analytical strategy
While conventional meta-analysis assumes that primary effects are independent, our data included correlated effects (i.e., multiple included effects from one primary participant sample). To account for this dependency across all analyses, we used a three-level meta-analytical model with inverse variance weights, where Level 1 corresponded with variation between participants, Level 2 with the variation of effect within each independent sample, and Level 3 with the variation of effects between independent participant samples[46,47]. All analyses were conducted using the metafor 4.6 package for R[48]. Figures were created with packages mapdata v.2.3.1[49] (Fig. 1) and ggplot2 v.3.5.2[50] (Figs. 2–8).

### Risk of bias and sensitivity analyses
Risk of bias assessment was based on the JBI tool for observational data[51]. Within each category of factors, we conducted a series of sensitivity analyses by excluding studies that were negatively evaluated on

any of the criteria of risk of bias assessment, unpublished studies, and outliers. Detailed results are presented in Supplementary Table 13. Overall, excluding studies with increased risk of bias did not affect the results in a noteworthy manner and did not change our conclusions[40].

**Publication bias and *p*-hacking**

Publication bias was assessed using multilevel Egger regression[52,53] and two variants of PET-PEESE[54,55]. Detailed results are presented in Supplementary Tables 14–15, and funnel plots are presented in Supplementary Fig. 1.

For each main category of predictors (see Results), effect sizes were regressed on their standard errors (Egger regression[53], or PET model), and on either sample size[54] or sampling variance[55] (PEESE) in a three-level model. In the case of Egger regression, results suggested the presence of publication bias in the case of the associations of cultural distance with psychological adaptation, and exposure to social groups with psychological adaptation, which might indicate that these effects are inflated. In the case of PET-PEESE, meta-regression intercepts indicate the size of unbiased effects. If the intercept of the PET model test is non-significant at $\alpha = 0.10$, PET results should be interpreted; if the intercept of the PET test is significant, PEESE results should be interpreted[55]. In all analyses, the estimated unbiased effects were consisted with the results of the core analyses presented here, except for the variance-based PEESE[55] which yielded larger effects of all predictors for socio-cultural adaptation. Therefore, PET-PEESE analyses did not show evidence of noteworthy bias.

Additionally, we tested for bias resulting from p-hacking by fitting right-truncated meta-analysis (RTMA)[56] models for all main categories of adaptation correlates (Supplementary Table 16). RTMA accounts for *p*-hacking by imputing, via Bayesian methods, the underlying distribution of population effects in order to estimate a corrected effect size. While RTMA is currently the only fully validated *p*-hacking correction method, it assumes that effects are independent. Our data violate this assumption, thus the results should be taken with extreme caution. Indeed, in two cases (cultural distance for psychological adaptation, stressors for socio-cultural adaptation), anomalies in results suggested that RTMA estimates were invalid (see Supplementary Table 16 for details). For the remaining results, the comparison of the corrected effects to uncorrected ones suggested that our core analyses underestimated the true effect in two cases (social resources for psychological adaptation; cultural distance for socio-cultural adaptation, with $\Delta r = 0.15$ and $\Delta r = 0.35$, respectively), and overestimated them in four cases (exposure and stressors for psychological adaptation, exposure and resources for socio-cultural adaptation, with $\Delta r$ ranging from 0.03 to 0.11).

**Reporting summary**

Further information on research design is available in the Nature Portfolio Reporting Summary linked to this article.

## Data availability

The data generated in this study have been deposited in the Open Science Framework repository under accession code https://doi.org/10.17605/OSF.IO/MD3SZ[57].

## Code availability

The code generated in this study have been deposited in the Open Science Framework repository under accession code https://doi.org/10.17605/OSF.IO/MD3SZ[57].

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

## Acknowledgements
This research was funded by the Research Council of Norway, grant 325260 (KB).

## Author contributions
Conceptualization: K.B., J.R.K., and C.W., Methodology: K.B., D.H.V., M.W.L.C., R.T., N.B., E.v.d., K.L., Investigation: K.B., D.H.V., R.T., N.B., E.v.D., K.L., M.W.L.C., Visualization: K.B., D.H.V., R.T., Funding acquisition: K.B., Writing – original draft: K.B., Writing – review & editing: K.B., J.R.K., D.H.V., R.T., C.W., M.W.L.C., N.B., E.v.d., K.L.

## Funding

## Competing interests
The authors declare no competing interests.
