## [Transparent Peer Review file · Nature Communications]

A Meta-Analysis of Social and Contextual Correlates of Migrant Adaptation to Living in Receiving Societies

Corresponding Author: Dr Kinga Bierwiazzonek

Version 0:

Reviewer comments:

Reviewer #1

(Remarks to the Author)

The author team did an excellent job putting together all the data, processing them, and making them accessible. Congratulations on this, this article will be a very important cornerstone for the next years in research on acculturation.

DATA

It is great to see the data on OSF, together with the code. I have not checked all of it, but it is informative. The data file could itself include a variable index (e.g., in a different sheet) to unambiguously identify the columns. It also seems that the data file is the illustration and not (all of) the actual data file on which the analyses were run (or those that were produced as part of the analyses listed in the R file). Maybe I am missing something here.

OVERVIEW AND PROCEDURE

The overview is limited to references to studies that showcase what still needs to be done, painting a relatively one-sided picture of a bigger discussion in the field of acculturation research. It could include also overviews (specifically regarding the integration hypothesis) that provide – together with the ones that are mentioned – a more nuanced perspective (e.g., <https://doi.org/10.1111/bjop.12656>, <https://doi.org/10.1016/j.ijintrel.2021.12.003>, <https://doi.org/10.1016/j.ijintrel.2023.101897>). Specifically the section on integration on p.4/5 could be more informative in that regard, which is all the more important for a publication in Nature Communications which will also reach readers not familiar with the conversation.

I understand that it is important to clarify how the current meta-analysis tackles things other studies, including meta-analyses have not been doing (the sheer amount of data is already a convincing argument, of course). There are a number of things I find somewhat problematic in the presentation, for instance is that other meta-analyses are listed for aspects/variables that were missed, but the current meta-analysis does not incorporate distinctions that these meta-analyses *do* cover (and presents itself as comprehensive). For instance, typologies of social support (ref 21 or this one, which is not included in the review: <https://doi.org/10.1016/j.hrmr.2018.12.003>; but it is in the supplemental section), or health-related acculturation outcomes (not included: <https://osf.io/preprints/psyarxiv/8qymv>). Health is also absent in the theoretical approach to the topic, as arguably one of the most important (and underresearched) elements of acculturation outcomes.

It would also be useful to clarify a bit what the different scientific fields are that are active in research on acculturation (Cultural studies, anthropology, business studies, etc.)

In the abstract, the authors refer to (perceived) cultural distance, and findings being limited; it is also mentioned as part of the culture-learning factors (p.6). This is non-trivial given that there are several objective indices, ranging from the the Kogut and Singh national cultural distance index (which is the one that is used), or the more recent operationalization of Muthukrishna, van de Vliert's latitudinal and climate approach, as well as others. Using the term perceived cultural distance suggests that there were self-report scales available for each of the effect sizes, likely based on the work by van de Vijver (<https://doi.org/10.1002/casp.989>, not mentioned). This variable is under-reported in the main text, and the details on p.19/20 could be elaborated (the pre-reg is also not clear).

I appreciate that, in general, more detailed information can be found in the supplemental material p.149, but the concepts/variables and their assessment as such are NOT presented there either. There is no list of measures, no reporting of how the data came about (beyond the search description, which is detailed). In my view, this is an important omission that needs to be addressed, specifically when heterogeneity is mentioned, as well as interdisciplinary differences in practices. The pre-registration (<https://osf.io/8zq2d>) has lots of useful information, but it is not always clear whether the final versions are the same.

RESULTS

With this in mind, I have difficulties interpreting the results (without consideration of instruments and conceptual clarity). The

results – taken by themselves – are very informative, and I do not see substantial issues (although I am not sure what would come up in case of having more information).

I think it would also be useful to account for some demographic aspects of the samples (age, gender), which are currently not considered/presented. Similarly it would be useful to understand where the samples are from, and what the geographic/cultural representation is. This is important as the authors suggest considerable coverage.

Related to that last point: not finding differences between the four different groups across contexts could also point toward little variability in terms of receiving contexts.

DISCUSSION

In terms of heterogeneity, it would be interesting to understand how much the disciplinary spread and specific practices in the intercultural field are related to this

MINOR

- Would it make sense to adopt a different color pattern than red-green (I am not sure whether red/green blindness warrants that, maybe the journal can weigh in here).

(Remarks on code availability)

please see above.

Reviewer #2

(Remarks to the Author)

I have carefully evaluated your manuscript entitled “Social and Contextual Correlates of Migrant Adaptation to Living in Receiving Societies: A Meta-Analysis of 1,114 Studies”. Overall, the manuscript presents a meta-analysis on factors affecting minority groups’ adaptation that offers a valuable contribution to acculturation research. The comprehensive synthesis of existing studies and the innovative approach to aggregating diverse factors are commendable. However, I have identified several concerns that require clarification and revision before the manuscript can be considered for publication in *Nature Communications*.

First, it would be informative to detail the countries where the included studies were conducted. Given that the majority of acculturation research has been undertaken in immigration societies, the manuscript should elaborate on the geographic concentration of the sampled regions. Although every society comprises minority groups undergoing acculturation—even if in relatively smaller numbers—a discussion delineating the specific countries/regions and cultural groups involved could significantly illuminate the extent to which the findings are generalizable.

The manuscript notes inconsistent results and replication challenges (p. 3). In relation to my concern above, these issues may, in part, be attributable to the heterogeneous nature of the ethnic groups and the differing socio-cultural contexts examined across studies. The discussion may be expanded to consider how variations in context and cultural specificity might influence the observed outcomes, thereby affecting both the generalizability and interpretation of the results.

Regarding the categorization of cultural distance as a culture-learning factor (p. 6), further clarification is warranted. The utilization of the cultural distance index as proposed by Kogut and Singh (1988) raises concerns, given its dated nature as an objective criterion. Whereas culture is assumed stable, it also changes and does so more rapidly in this modernized society (Inglehart & Welzel, 2005). To look at the list of the studies examined, most of them were conducted within the last decade (2015~). It is important to specify whether the examined studies have employed additional or alternative criteria (p. 19) and, if so, to provide a clear description of these measures. Such clarification would enhance the validity of the categorization.

The manuscript posits a focus on the relative importance of stress-and-coping factors (i.e., stressors and social resources) and culture-learning factors (i.e., social interaction/exposure and cultural distance). However, the concept of culture-learning is ambiguous; it is not self-evident that frequent interactions with various groups automatically facilitate cultural learning, nor that perceived closeness or distance consistently exerts a beneficial influence. A reconsideration of the label—potentially simplifying it to “exposure and distance (or difference)” —may provide greater conceptual clarity. Additionally, the order of sub-categories appears inconsistent in Supplementary Tables.

Finally, it would be beneficial to clearly define “cultural distance” within the manuscript. If the term is intended to denote the differences (as opposed to similarities) between the heritage and host societies, this should be explicitly stated. Reconsideration of the label may help avoid conceptual ambiguity and ensure that readers fully grasp the intended construct.

Inglehart, R., & Welzel, C. (2005). *Modernization, Cultural Change, and Democracy: The Human Development Sequence*. Cambridge: Cambridge University Press.

(Remarks on code availability)

Reviewer #3

(Remarks to the Author)

The paper provides a quantitative synthesis of the extensive research on factors influencing migrant adaptation. The authors compile data from more than 1,000 studies, covering over half a million migrants. Their findings suggest that the most significant factors associated with adaptation are perceived discrimination, social resources, and feelings of connectedness. I believe this meta-analysis is highly valuable and has the potential to be widely influential. I would like to see it in print. However, in its current form, the analysis presented in the paper is problematic, and major revisions are necessary to justify publication in a top journal such as Nature Communications. My comments follow.

1) Publication bias and p-hacking

My primary concern is that the paper does not adjust for publication bias and p-hacking. The authors briefly discuss publication bias (p. 30) and conduct basic tests using Egger regression. They even detect bias, stating:

"Results suggested the presence of publication bias in the associations of cultural distance with psychological adaptation and exposure to social groups with psychological adaptation, which might indicate that these effects are inflated."

Yet, no correction is applied to the results. Given that publication bias can exaggerate reported estimates by a factor of two or more, adjusting for it in meta-analysis is essential.

The meta-analysis literature offers multiple correction methods, broadly classified into selection models and funnel-plot-based models. Given the richness and complexity of the data in this study, both types should be applied. I encourage the authors to implement at least one method from each category: for example, the PET-PEESE model (a simple yet robust funnel-based approach) and the 3PSM model (a frequently used selection model).

Additionally, p-hacking, which is distinct from publication bias (see work by Maya Mathur on this topic), should be addressed. Currently, two existing techniques can correct for both p-hacking and publication bias: RTMA and MAIVE. The authors should consider applying at least one of these methods. Given the potential for inflated effect sizes, correcting for these biases could significantly alter the study's conclusions.

2) Meta-analysis methods

The paper does not clearly specify which meta-analysis technique is used. Is it the standard random-effects model with inverse-variance weighting? Fixed effects? Are weights based on sample size or not applied at all?

This omission is critical because in meta-analyses of observational studies, reported precision can be misleading. If inverse-variance weights are used without adjusting for spurious precision, the estimates may be biased. The supplementary information on methodology requires significantly more detail, and key methodological choices should be outlined in the main text as well.

3) Causality

The meta-analysis synthesizes correlations, not causal effects. The authors acknowledge this on p. 6, yet their language throughout the paper often implies causality. For instance:

Abstract: "Understanding what factors contribute to the successful and equitable inclusion of migrants is a matter of highest urgency. Our meta-analysis helps pinpoint such factors."

Page 5: "Important antecedents of adaptation," "Factors crucial to migrant adaptation."

The authors must be much more cautious with causal language. If they wish to discuss causality (which I encourage), they should focus on a subset of quasi-experimental studies that explicitly identify or try to identify causal effects. Furthermore, the risk-of-bias analysis is far more useful for causal inference than for correlational research, and I would also like to see more details on this front.

4) Heterogeneity

The authors do acknowledge heterogeneity, but its levels are extreme (I2 approaching 100% in some cases). I find it problematic to pool migrants, refugees, business expatriates, and international students together. While the authors do report separate meta-analyses for these groups, if they wish to combine estimates further, a more logical grouping would be:

Students and expatriates together
Migrants and refugees together

Pooling all groups into a single estimate is not meaningful given the vast differences in migration experiences (and wealth).

5) Migrant characteristics

The paper would be much stronger if it examined how migrant characteristics (e.g., legal vs. undocumented status, country of origin, socioeconomic background) influence adaptation. The authors likely have access to relevant data for at least some of these variables. Including such an analysis would greatly enhance the study's impact.

6) Style

The paper is competently written, but some wording could be simplified to improve accessibility. Additionally, the authors repeatedly emphasize that their research addresses a topic of high societal relevance. While true, these repeated statements could be trimmed for conciseness.

The introduction to the Results section is difficult to follow. For example, the phrase "intercept-only models" will be unclear to many readers.

The results are presented monotonously, listing numbers for different categories without sufficient context or comparisons. Restructuring this section to highlight key contrasts would improve readability.

7) Tables and figures

All tables and figures should be self-explanatory, but some are not. For example, Table 1 is difficult to interpret. Readers struggle to understand what each row represents. More informative captions, notes, and clearer labeling would improve clarity.

Despite these issues, I want to emphasize that this meta-analysis is fundamentally well-grounded: particularly in terms of data collection and literature search. A significant amount of effort has been invested in this paper, and I strongly believe that a revised version will be an extremely valuable contribution to both academia and policy discussions. With the necessary corrections, this study has the potential to become a highly cited, influential piece of research.

(Remarks on code availability)

Version 1:

Reviewer comments:

Reviewer #1

(Remarks to the Author)

The authors have done a very good job in addressing the comments raised by me earlier. I appreciate the limitations that the authors experienced in navigating the format of the journal (and taking a moment to also add this in the cover letter, very helpful to highlight). I appreciate the additional details (specifically the heatmap, Figure 1, is an excellent addition).

I disagree with the assessment of the authors that a further discussion of heterogeneity as a result of disciplines may not be relevant for the (broader) readership, as this is an issue that may apply to numerous other fields too. The authors state that the methodology and measures are fairly similar, but I find that difficult to follow, given that even within the cultural field there are different standards of reporting (reliabilities, translation/adaptation, etc.). I had no access to the chapter referenced.

Looking forward to seeing the manuscript in print.

Minor:

I think it would be useful to rephrase “the Global North seemed overrepresented in our data” to “the Global North is overrepresented in our data” and “This geographical imbalance may limit the generalizability of our findings” to “This geographical imbalance limits the generalizability of our findings”.

The section on the underrepresentation of the Global South transitions to the call for multi-level analyses (which I agree with), but I think the link does not work out as well, as this is more an issue of the number of studies needed.

(Remarks on code availability)

I have not extensively checked the code, but have spot-checked elements of it.

Reviewer #2

(Remarks to the Author)

I appreciate the authors' efforts in revising the manuscript. The revised version has improved in several respects. However, several issues remain that require further clarification and revision to enhance the clarity and rigor of the study:

The reported average effects of social resources appear to be misleading. In Supplementary Table 1, social resources do not show significant effects overall. This discrepancy may be due to the fact that significant effects were observed only for specific sources of support—namely, from supervisors (p. 15). Please revise the relevant sections to reflect this nuance accurately and correct the potentially misleading interpretation.

The term acculturative stressors is described as an umbrella term (p. 13), yet no clear definition or operationalization is provided in the Method section, as the authors suggest. This is problematic, especially because the constructs commonly considered acculturative stressors (e.g., language barriers, discrimination) appear to be analyzed separately in the current study. For example, the Acculturative Stress Index (Noh et al.) used in the analysis (ST 12) includes domains such as language barriers, social isolation, discrimination, and lack of occupational opportunities—all of which are treated as distinct variables in this study. This overlap raises concerns about conceptual clarity and potential redundancy. A clearer explanation of how items were selected and categorized is necessary, as the current reporting may mislead readers.

To strengthen the practical implications of the findings, and to avoid conceptual overlap similar to the issue above, the manuscript would benefit from including example items related to support from authority figures. While such support is shown to be beneficial, it remains unclear what kinds of support are most effective. At minimum, the authors could discuss whether the items reflect relationship-oriented support (e.g., emotional encouragement) or task-oriented support (e.g.,

administrative assistance). Clarifying the nature of this support would enhance the real-world relevance of the findings for organizational settings.

The relatively weak or non-significant effects of both perceived and objective cultural distance warrant more careful interpretation. It is surprising that cultural distance does not emerge as an important factor in the experiences of migrants and refugees. Readers would benefit from a clearer description of how cultural distance was measured, including how the relevant questions were presented to participants. It would be particularly helpful to list the full set of items used in each scale. If this is not feasible, providing representative example items would still add clarity. While the authors briefly consider the possibility of subgroup variation (p. 24), further discussion on the potential measurement limitations and the need for more robust assessment would strengthen the manuscript's contribution.

(Remarks on code availability)

Reviewer #3

(Remarks to the Author)

The authors have successfully incorporated my suggestions, as well as those from the other referees. The revised manuscript is now much clearer, more transparent, and includes appropriate publication bias corrections using both PET-PEESE and a robust Bayesian model. My only remaining concern is the absence of any method that addresses p-hacking. This is particularly relevant in a meta-analysis of psychological studies, where selective reporting of significant results is a well-documented problem.

Methods such as RTMA or MAIVE are straightforward to implement and suitable for robustness checks, even in the presence of heterogeneity and dependent effect sizes. For example, working MAIVE code is available at https://meta-analysis.cz/maive/maive_R_package.zip. If they have any troubles running it, they can use any general IV package available in R or Stata. Of course, as the authors rightly point out, these methods rely on specific assumptions, but those can and should be explicitly discussed. Many papers have shown that p-hacking is important in observational research. Including at least one p-hacking robust estimator, even as a secondary analysis, would substantially strengthen the credibility of the findings and align the paper even more closely with current best practices in meta-research.

(Remarks on code availability)

Version 2:

Reviewer comments:

Reviewer #1

(Remarks to the Author)

I do not have further comments that require action and would like to thank the authors for the very nice job of incorporating the feedback I raised.

Just in case it is helpful to the authors: I still cannot access the OSF link associated with the Vu et al in press paper (https://osf.io/sk6mj_v1?view_only=)

Page returns: "Page not found"

The requested resource could not be found. If this should not have occurred and the issue persists, please report it to support@osf.io."

(Remarks on code availability)

I have not gone through the previously reviewed files again, so clicked on "not having reviewed the code" above.

Reviewer #2

(Remarks to the Author)

I have reviewed all the materials submitted after the revision. I find that my previous comments have been thoroughly addressed and well reflected in the revised documents. I am pleased to recommend the manuscript for acceptance and publication.

(Remarks on code availability)

Reviewer #3

(Remarks to the Author)

The authors have improved the paper again and incorporated my additional comments. I believe the paper is now ready for publication.

(Remarks on code availability)

RESPONSE TO REVIEWER COMMENTS

Reviewer #1

R1.1. *The author team did an excellent job putting together all the data, processing them, and making them accessible. Congratulations on this, this article will be a very important cornerstone for the next years in research on acculturation.*

Response to R1.1. We thank the Reviewer for their positive feedback.

R1.2 DATA

It is great to see the data on OSF, together with the code. I have not checked all of it, but it is informative. The data file could itself include a variable index (e.g., in a different sheet) to unambiguously identify the columns.

Response to R1.2. We thank the Reviewer for this suggestion. Accordingly, we have now added a variable index to the datafile.

R1.3 *It also seems that the data file is the illustration and not (all of) the actual data file on which the analyses were run (or those that were produced as part of the analyses listed in the R file). Maybe I am missing something here.*

Response to R1.3. We would like to clarify that the dataset and code published on OSF is complete and covers all the analyses reported in this manuscript. Please note that most variables and subsets are not visible in the dataset because they were created using the R code.

R1.4. OVERVIEW AND PROCEDURE

The overview is limited to references to studies that showcase what still needs to be done, painting a relatively one-sided picture of a bigger discussion in the field of acculturation research. It could include also overviews (specifically regarding the integration hypothesis) that provide – together with the ones that are mentioned – a more nuanced perspective (e.g., <https://doi.org/10.1111/bjop.12656>, <https://doi.org/10.1016/j.ijintrel.2021.12.003>, <https://doi.org/10.1016/j.ijintrel.2023.101897>). Specifically the section on integration on p.4/5 could be more informative in that regard, which is all the more important for a publication in Nature Communications which will also reach readers not familiar with the conversation.

Response to R1.4. In line with the Reviewer's comment, we have now cited the listed references in the Introduction (pp. 4-5). We have also reworded the relevant passage in the Introduction that currently reads:

While research on these correlates is extensive, empirical support for their relevance is inconsistent, and some of them may not be backed by solid evidence. Perhaps the most prominent example is the acculturation strategy of integration (i.e., an orientation toward both the migrant's heritage culture and the receiving culture). Long seen as a critical predictor of good adaptation^{2,17,20}, integration was recently revealed to have modest and extremely heterogeneous associations with adaptation^{14,15}, triggering an ongoing debate on the practical relevance of this correlate^{16,17,18, 21}. This underscores the necessity of examining how robust the correlates put forward by the broader literature really are.

Moreover, we would like to clarify that, since acculturation strategies are not the main focus of the current paper, we refrained from an extensive discussion of the complex issue of associations between integration and adaptation; first, in order to avoid distracting the reader from the main focus of this paper, that is, social and contextual correlates of adaptation; second, because the issue is well captured in the various cited papers; third, because we would not be able to do it justice within *Nature Communication's* word limit.

R1.5. *I understand that it is important to clarify how the current meta-analysis tackles things other studies, including meta-analyses have not been doing (the sheer amount of data is already a convincing argument, of course). There are a number of things I find somewhat problematic in the presentation, for instance is that other meta-analyses are listed for aspects/variables that were missed, but the current meta-analysis does not incorporate distinctions that these meta-analyses *do* cover (and presents itself as comprehensive). For instance, typologies of social support (ref 21 or this one, which is not included in the review: <https://doi.org/10.1016/j.hrmr.2018.12.003>; but it is in the supplemental section), or health-related acculturation outcomes (not included: <https://osf.io/preprints/psyarxiv/8qymv>).*

Response to R1.5. We thank the Reviewer for this comment and for suggesting moving van der Laken et al. (2019) to the main manuscript. We have now brought this reference back and we have rewritten the relevant passage in the introduction, clarifying that our mention of the previous meta-analyses' defined scope is not intended as criticism, but rather serves to identify the existing coverage and gaps in the field (p. 5).

Despite offering focused and critical insights on specific factors relevant to adaptation, previous meta-analyses did not attempt to integrate this vast and dispersed literature due to their specialized foci. Specifically, one meta-analysis only covered correlates of one specific measure of adaptation (the Socio-Cultural Adaptation Scale)¹⁹, two meta-analyses covered only one measure of adjustment for one specific migrant population (a scale of adjustment for expatriate managers)^{7,8}, and two other meta-analyses covered in great detail one variable for one migrant population (social support for international students²² and for business expatriates²⁴).

Additionally, inspired by the abovementioned meta-analysis, we have now added an analysis differentiating the types of social support to the supplemental information file, Supplementary

Tables 10-11. Please note as well Figures 6 and Supplementary Tables 8-9 that report the different sources of social resources, which closely resemble the typologies covered by van der Laken et al. (2019).

Finally, we would like to clarify that we had to prioritize the literature cited in the main manuscript because *Nature Communications* only allows for a limited number of references. Therefore, as a rule we refrained from citing works that had not been peer-reviewed, which is the case for the second link provided by the Reviewer in this comment.

RI.6. Health is also absent in the theoretical approach to the topic, as arguably one of the most important (and underresearched) elements of acculturation outcomes.

Response to RI.6. Indeed, the theoretical framework for this specific manuscript builds on Colleen Ward's seminal and highly influential work that is concerned specifically with psychological and socio-cultural adaptation and does not distinguish health-related outcomes as a separate adaptation outcome. The same two-dimensional approach has been the organizational, framework used in systematic reviews (e.g., Bierwiazzonek & Waldzus, 2016, ref. 13), bibliometric analyses (e.g., Tang & Zhang, 2023) and meta-analyses (e.g., Grigoryev et al., 2022). We thus selected it, in part, for the sake of consistency with the literature.

This said, we agree with the Reviewer that health outcomes are important and deserve proper consideration, something we now explicitly discuss on p. 28:

Moreover, our analysis focused on the most influential framework within acculturation research, which encompasses psychological and sociocultural dimensions of adaptation. However, physical health remains a significantly understudied and undertheorized dimension of acculturation. Future theories, primary work, and meta-analyses could benefit from including physical health as an additional outcome.

References:

Tang, L., & Zhang, C. A. (2023). Global research on international students' intercultural adaptation in a foreign context: A visualized bibliometric analysis of the scientific landscape. *SAGE Open*, 13(4), 21582440231218849.
Grigoryev, D., Berry, J. W., Stogianni, M., Nguyen A.-M.T. D., Bender, M., & Benet-Martínez, V. (2023). The integration hypothesis: A critical evaluation informed by multilevel meta-analyses of three multinational datasets. *International Journal of Intercultural Relations*, 97(6), 1–14.

RI.7. It would also be useful to clarify a bit what the different scientific fields are that are active in research on acculturation (Cultural studies, anthropology, business studies, etc.)

Response to RI.7. We thank the Reviewer for this suggestion. On p. 3, we now list some of the main fields of studies:

Given these social outcomes, the increase of worldwide migration has been accompanied by an exponential growth of studies on migrant acculturation and adaptation across various fields such as psychology, cultural studies and anthropology, organizational studies, communication studies, and sociology.

Please note, however, that we do not have an exhaustive list of active scientific fields because many disciplines (e.g., ethnography, human geography) conduct studies with migrants with methods that drastically differ from psychological methods and thus are not eligible for a meta-analysis like ours.

RI.8. *In the abstract, the authors refer to (perceived) cultural distance, and findings being limited; it is also mentioned as part of the culture-learning factors (p.6). This is non-trivial given that there are several objective indices, ranging from the Kogut and Singh national cultural distance index (which is the one that is used), or the more recent operationalization of Muthukrishna, van de Vliert's latitudinal and climate approach, as well as others. Using the term perceived cultural distance suggests that there were self-report scales available for each of the effect sizes, likely based on the work by van de Vijver (<https://doi.org/10.1002/casp.989>, not mentioned). This variable is under-reported in the main text, and the details on p.19/20 could be elaborated (the pre-reg is also not clear).*

Response to RI.8. We thank the Reviewer for this comment. As preregistered, we coded measures of cultural distance whenever these were made available in the included body of research (we did not calculate cultural distance ourselves for any of the included studies). As such, whether this measure referred to self-reported cultural distance or objectively measured cultural distance depended on the respective studies. The same applies to the choice of objective measures: These reflect the choice of the authors of the included work rather than our own preference of one over the other available indices.

Considering that the included measures were both self-reported and based on objective criteria, we agree with the Reviewer that the term “perceived cultural distance” was misleading. We thus removed the word “perceived” from the abstract/discussion and now refer simply to “cultural distance.” Moreover, as suggested, we added clarifications on p. 23:

We compared the effects of self-rated cultural distance with the effects of objectively measured cultural distance whenever available in the reviewed studies. Self-rated distance referred to perceived differences between the heritage culture and the receiving culture as reported by participants themselves. Objectively measured distance referred to indexes tapping into cultural differences between two given countries based on external criteria, such as differences in cultural values from external databases (e.g., Kogut & Singh's index²⁵). In all cases, the specific indexes were selected and applied by the authors of the primary studies.

RI.9. *I appreciate that, in general, more detailed information can be found in the supplemental material p.149, but the concepts/variables and their assessment as such are NOT presented there*

either. There is no list of measures, no reporting of how the data came about (beyond the search description, which is detailed). In my view, this is an important omission that needs to be addressed, specifically when heterogeneity is mentioned, as well as interdisciplinary differences in practices.

Response to R1.9. We thank the Reviewer for this inquiry. As they noted in the following comment, the previous version of the paper linked to the full protocol with a detailed coding scheme explaining the used constructs under Data Items. However, we agree that the accessibility of this information could be improved. Therefore, we now provide the requested list of covered constructs with measures in the supplemental information, specifically Supplementary Table 12.

R1.10. *The pre-registration (<https://osf.io/8zq2d>) has lots of useful information, but it is not always clear whether the final versions are the same.*

Response to R1.10. Please note that, in line with standard procedures, all deviations from the plan were noted in Version 2 of the preregistration protocol, under Section 4: Additional information, as preregistered in the original protocol. No other changes were made. We provided the link to the registered deviations on p. 29 of the main manuscript:

The analyses presented here are part of Work Package 2 in the preregistration protocol³⁹ (see also https://osf.io/qc9h2/resources?mode=&revisionId=&view_only= for registered deviations from the plan).

R1.11. RESULTS

With this in mind, I have difficulties interpreting the results (without consideration of instruments and conceptual clarity). The results – taken by themselves – are very informative, and I do not see substantial issues (although I am not sure what would come up in case of having more information).

Response to R1.11. We thank the Reviewer for the overall positive feedback, and we hope that the additional information provided in response to R1.9 and R1.10 facilitates comprehension.

R1.12. *I think it would also be useful to account for some demographic aspects of the samples (age, gender), which are currently not considered/presented.*

Response R1.12. We thank the Reviewer for this suggestion. Detailed information about sample socio-demographics is reported in Table 1, which we have now revised to be clearer.

In addition, for interested readers, we have added meta-regressions testing for sample gender and age differences to the supplementary materials, see Supplementary Table 7. This analysis

Editorial Note: Figure 1 on this page of the Transparent Peer Review file was created using the `map_data` function from the R package `mapdata` v.2.3.1.⁴⁹

tentatively suggests that older samples might be less affected by stressors and cultural distance, but we refrained from discussing this result for two reasons. First, sample age is highly correlated with length of stay, a variable that we chose not to use in analyses because of high percentage of missing data. Interpreting this result might therefore be misleading. Second, we do not consider sample-level socio-demographics to be particularly informative as a moderator, as they are prone to misinterpretations and ecological fallacies (i.e., inferring individual-level effects from sample means; see, for example, Geissbühler et al., 2021).

Reference:

Geissbühler, M., Hincapié, C. A., Aghlmandi, S., Zwahlen, M., Jüni, P., & da Costa, B. R. (2021). Most published meta-regression analyses based on aggregate data suffer from methodological pitfalls: A meta-epidemiological study. *BMC Medical Research Methodology*, 21(1), 123.

R1.13. Similarly it would be useful to understand where the samples are from, and what the geographic/cultural representation is. This is important as the authors suggest considerable coverage.

Response to R1.13. In line with the Reviewer's comment, we have now added heatmaps that illustrate which countries participants came from and which they resided in, see Figure 1, added here for the Reviewer's convenience.

Figure 1.

Geographical coverage of the meta-analysis. The number of participants per country of origin is presented in the top panel, and the number of participants per receiving country is presented in the bottom panel. Please note that this figure excludes samples with mixed or unknown origins/receiving countries.

***R1.14.** Related to that last point: not finding differences between the four different groups across contexts could also point toward little variability in terms of receiving contexts.*

Response to R.14. We thank the Reviewer for this suggestion. We would argue that, on the contrary, genuine differences between groups should be even clearer with less variability (i.e., keeping reception context constant). Moreover, as per Figure 1, our dataset includes a variety of receiving contexts, and in our view the fact that we still do not find differences between groups strengthens the argument that these effects apply across groups.

R1.15. DISCUSSION

In terms of heterogeneity, it would be interesting to understand how much the disciplinary spread and specific practices in the intercultural field are related to this

Response to R1.15. We agree with the Reviewer that this is an interesting question, however, it may be mostly relevant for specialists in the intercultural area, while Nature journals aim at a broader readership. We also believe that such a comparison has relatively limited value given that this meta-analysis relied on studies using the same type of methodology and measures, which limits potential interdisciplinary variability that could be observed.

Please note as well that we have already provided a discussion of field practices in the book chapter referenced below:

Vu, D.H., Kunst, J.R., Tong, R., & Bierwiazzonek, K. (*in press*). Methodological challenges in cross-cultural adaptation research: Insights from a large-scale meta-analysis. In: L. A. Arasaratnam-Smith (Ed.). *Handbook of Communication and Culture*. Edward Elgar Publishing. https://doi.org/10.31234/osf.io/sk6mj_v1

R1.16. MINOR

- Would it make sense to adopt a different color pattern than red-green (I am not sure whether red/green blindness warrants that, maybe the journal can weigh in here).

Response to R1.16. We thank the Reviewer for drawing our attention to this point. The colors were taken from a color scheme specifically designed for color-blind people, but we agree with the Reviewer that even so, the choice of green and red was perhaps unfortunate from this perspective. We have now changed the colors as suggested to blue and pink.

Reviewer #2

R2.1. *I have carefully evaluated your manuscript entitled “Social and Contextual Correlates of Migrant Adaptation to Living in Receiving Societies: A Meta-Analysis of 1,114 Studies”. Overall, the manuscript presents a meta-analysis on factors affecting minority groups’ adaptation that offers a valuable contribution to acculturation research. The comprehensive synthesis of existing studies and the innovative approach to aggregating diverse factors are commendable.*

Response to R2.1. We thank the Reviewer for their positive and constructive feedback.

R2.2. *However, I have identified several concerns that require clarification and revision before the manuscript can be considered for publication in Nature Communications.*

First, it would be informative to detail the countries where the included studies were conducted. Given that the majority of acculturation research has been undertaken in immigration societies, the manuscript should elaborate on the geographic concentration of the sampled regions. Although every society comprises minority groups undergoing acculturation—even if in relatively smaller numbers—a discussion delineating the specific countries/regions and cultural groups involved could significantly illuminate the extent to which the findings are generalizable.

The manuscript notes inconsistent results and replication challenges (p. 3). In relation to my concern above, these issues may, in part, be attributable to the heterogeneous nature of the ethnic groups and the differing socio-cultural contexts examined across studies. The discussion may be expanded to consider how variations in context and cultural specificity might influence the observed outcomes, thereby affecting both the generalizability and interpretation of the results.

Response to R2.2. We thank the Reviewer for this suggestion. We have now added heatmaps that illustrate which countries participants come from and which they resided in, see Figure 1.

Furthermore, we now discuss the consequences of the observed migration patterns for the generalizability of results on p. 28:

When interpreting the results of the present meta-analysis, it is important to acknowledge that despite the diverse range of origin and reception contexts covered, the Global North seemed overrepresented in our data (see Figure 1). This geographical imbalance may limit the generalizability of our findings, for instance regarding migration trajectories within the Global South. Future research should address this limitation by directly examining the role of country-level factors through comprehensive multi-level analyses, which would provide more nuanced insights into potential contextual variations in acculturation processes.

Moreover, on p. 26, we now note that the diversity in contexts likely contributed to the observed heterogeneity (the citation refers to Bornstein, 2017):

Crucially, heterogeneity was extreme in all analyses, which was expected considering similarly high levels of heterogeneity in existing meta-analyses on acculturation^{15,16} and the diversity of cultural contexts of origin and reception covered by this study⁵⁴.

Reference:

Bornstein, M. H. (2017). The Specificity Principle in Acculturation Science. *Perspectives on Psychological Science*, 12(1), 3-45.

R2.3. *Regarding the categorization of cultural distance as a culture-learning factor (p. 6), further clarification is warranted.*

Response to R2.3. We agree with the Reviewer that our rationale for considering cultural distance as related to culture learning could be clarified further. In psychological literature, cultural distance is commonly categorized as a culture learning factor (see Wilson et al., 2013, ref. 19 in the manuscript) because it determines whether or not a behavior learnt within the culture of origin can be effectively used (i.e., generalized) to the culture of reception (i.e., learning generalization). Such a generalization is likely facilitated if cultures are relatively similar (i.e., low cultural distance) but more difficult if they are dissimilar (i.e., high cultural distance). Moreover, learning theory and research shows that “less familiar stimuli are more difficult to combine to create new knowledge and that this is because less familiar stimuli consume more working memory resources” (Reder et al., 2016), that is, less familiar stimuli are more difficult to learn. In other words, cultural distance can function as a proxy for the *amount* of learning required and the level of its *difficulty*, so that greater distance requires more effortful learning. By consequence, lower distance should be associated with better socio-cultural adaptation.

We have now added a clarification of this issue in the main manuscript, p. 4:

The culture learning process behind socio-cultural adaptation is believed to be influenced by factors related to social learning (i.e., learning via exposure to the receiving society, such as through social interactions with receiving country nationals, but also co-nationals or individuals from other migrant groups) and learning generalization (i.e., being able to generalize culture-specific behaviors and knowledge acquired in one context to another cultural context, which depends, among others, on the degree of cultural distance, that is, dissimilarity between the receiving and heritage cultures)^{11,19}.

Reference:

Reder, L. M., Liu, X. L., Keinath, A., & Popov, V. (2016). Building knowledge requires bricks, not sand: The critical role of familiar constituents in learning. *Psychonomic Bulletin & Review*, 23, 271-277.

R2.4. *The utilization of the cultural distance index as proposed by Kogut and Singh (1988) raises concerns, given its dated nature as an objective criterion. Whereas culture is assumed stable, it also changes and does so more rapidly in this modernized society (Inglehart & Welzel, 2005). To*

look at the list of the studies examined, most of them were conducted within the last decade (2015~). It is important to specify whether the examined studies have employed additional or alternative criteria (p. 19) and, if so, to provide a clear description of these measures. Such clarification would enhance the validity of the categorization.

Response to R2.4. We thank the Reviewer for raising this issue. Our lack of clarity regarding cultural distance has led to a misunderstanding, for which we apologize.

We would like to clarify that measures of cultural distance were not chosen by us; they were selected and applied by the authors of the primary studies that we included at the time when their study was conducted. Thus, we did not have any control over the fact that Kogut & Singh's index was more frequently used than newer, perhaps less problematic approaches. Please note as well that this index was cited as one example, but other measures were also used in the primary studies, albeit less frequently (e.g., Dow & Karunaratna, 2006).

Thus, in line with our pre-registration, we did not calculate cultural distance ourselves for any of the included studies. As usual for meta-analyses, we merely aggregated the correlations with adaptation outcomes reported by primary studies. To do so, we used all and any measures of cultural distance reported in the primary studies.

Please note that even if we had attempted to calculate “objective” cultural distance based on the information about the country of origin and the receiving country, this score would not have been useable for the kind of analyses we report here because it would be impossible for us to calculate within-study correlations of such recalculated cultural distance scores with adaptation outcomes. There are two reasons for that: (a) most studies used samples from a single origin residing in a single receiving country, thus the distance score would be a constant; (b) even if studies used mixed origin samples, we would not have had enough information to reliably calculate such correlations because we worked at a study level, not a participant level, and in an overwhelming majority of cases we did not have access to individual participant data.

As suggested, we added a clarification of this issue on p. 23:

We compared the effects of self-rated cultural distance with the effects of objectively measured cultural distance whenever available in the reviewed studies. Self-rated distance referred to perceived differences between the heritage culture and the receiving culture as reported by participants themselves. Objectively measured distance referred to indexes tapping into cultural differences between two given countries based on external criteria, such as differences in cultural values from external databases (e.g., Kogut & Singh's index²⁵). In all cases, the specific indexes were selected and applied by the authors of the primary studies.

R2.5. *The manuscript posits a focus on the relative importance of stress-and-coping factors (i.e., stressors and social resources) and culture-learning factors (i.e., social interaction/exposure and cultural distance). However, the concept of culture-learning is ambiguous; it is not self-evident that frequent interactions with various groups automatically facilitate cultural learning, nor that*

perceived closeness or distance consistently exerts a beneficial influence. A reconsideration of the label—potentially simplifying it to “exposure and distance (or difference)”—may provide greater conceptual clarity. Additionally, the order of sub-categories appears inconsistent in Supplementary Tables.

Response to R2.5. We thank the Reviewer for this suggestion. In the revised manuscript, we have simplified the labels as suggested. We also agree with (and the data seem to support) the point that it is not self-evident that frequent interactions with various groups automatically facilitate cultural learning. However, this is a theoretical assumption put forward in the literature on cross-cultural adaptation, based mostly on Bandura’s Social Learning Theory, which emphasizes the importance of observing, modeling, and imitating the behaviors of others. Thus, in an intercultural context, exposure to and interactions with individuals in the receiving society provide behaviors to be modelled. This theoretical assumption was among those our meta-analysis set out to test.

Given the above, we decided to maintain the concept of culture learning as part of the theoretical framing in the Introduction and Discussion, but we attempted to be clear that its relation to the tested variables is theoretical rather than empirical, for instance:

*The culture learning process behind socio-cultural adaptation **is believed** to be influenced by factors related to social learning (...) and learning generalization (...). (p. 4).*

*Specifically, based on the frameworks reviewed so far, we focus on the relative importance of (a) social resources and stressors (both general and specific to migration, i.e., acculturative stressors), as factors theorized to be related to stress-and-coping, and (b) exposure to social groups within the receiving culture and cultural distance between the culture of origin and the receiving culture, as factors **theorized to** be related to culture learning. (p. 6)*

*The role of variables **theorized to** be related to culture learning, namely exposure to social groups that can model or explain the new culture, and the cultural distance between the new culture and one’s heritage culture (...). (p. 23)*

Additionally, we have now ordered all main categories of correlates in the Supplementary Tables so that they match the order in which they are discussed in the manuscript. However, we would like to clarify that for meta-regressions testing correlate subcategories (Supplementary Tables 8-9), a consistent order is not always possible. The subcategories in the Figures are ordered from the smallest effect to the largest for readability, whereas the meta-regressions’ order depends on the baseline subcategory, which needs to be a meaningful one (in most cases, the largest effect size). By consequence, the order is consistent with the Figures in the case of stressors and cultural distance (as the effects are negative), but not in the case of resources and exposure because for those categories, the baseline does not match the first effect displayed in the figure.

R2.6. Finally, it would be beneficial to clearly define “cultural distance” within the manuscript. If the term is intended to denote the differences (as opposed to similarities) between the heritage and host societies, this should be explicitly stated. Reconsideration of the label may help avoid conceptual ambiguity and ensure that readers fully grasp the intended construct.

Response to R2.6. We thank the Reviewer for this suggestion. We considered changing the label to “cultural difference,” but the bulk of psychological literature on migration synthesized here uses the label “cultural distance.” We thus opted for maintaining this label for the sake of consistency with the primary literature. However, as suggested, we now clearly define cultural distance as

dissimilarity between the receiving and heritage cultures (p. 4).

Reviewer #3

R3.1. *The paper provides a quantitative synthesis of the extensive research on factors influencing migrant adaptation. The authors compile data from more than 1,000 studies, covering over half a million migrants. Their findings suggest that the most significant factors associated with adaptation are perceived discrimination, social resources, and feelings of connectedness. I believe this meta-analysis is highly valuable and has the potential to be widely influential. I would like to see it in print.*

Response to R3.1. We thank the Reviewer for this positive evaluation.

R3.2. *However, in its current form, the analysis presented in the paper is problematic, and major revisions are necessary to justify publication in a top journal such as Nature Communications. My comments follow.*

1) Publication bias and p-hacking

My primary concern is that the paper does not adjust for publication bias and p-hacking. The authors briefly discuss publication bias (p. 30) and conduct basic tests using Egger regression. They even detect bias, stating:

"Results suggested the presence of publication bias in the associations of cultural distance with psychological adaptation and exposure to social groups with psychological adaptation, which might indicate that these effects are inflated."

Yet, no correction is applied to the results. Given that publication bias can exaggerate reported estimates by a factor of two or more, adjusting for it in meta-analysis is essential.

Response to R3.2. We thank the Reviewer for raising this issue. However, addressing publication bias and p-hacking requires robust assumptions regarding how the data are generated and reported. For instance, corrections for publication bias and p-hacking typically assume that effect sizes are independent and with small amount of heterogeneity (see, for example, Stanley, 2017). Violating these assumptions can significantly affect the corrections applied, making it challenging to determine whether the corrected values are adequate. Therefore, we refrained from presenting corrected results as our core results. However, we have implemented some of the methods recommended by the Reviewer, and we explain how in the following responses (see R3.3, R3.4, and R3.5).

Reference:

Stanley, T. D. (2017). Limitations of PET-PEESE and Other Meta-Analysis Methods. *Social Psychological and Personality Science*, 8(5), 581-591. <https://doi.org/10.1177/1948550617693062>

R3.3. *The meta-analysis literature offers multiple correction methods, broadly classified into selection models and funnel-plot-based models. Given the richness and complexity of the data in this study, both types should be applied. I encourage the authors to implement at least one*

method from each category: for example, the PET-PEESE model (a simple yet robust funnel-based approach) and the 3PSM model (a frequently used selection model).

Response to R3.3. We thank the Reviewer for these recommendations. We refrained from using 3PSM because Rodgers and Pustejovsky (2021) demonstrated that the 3PSM does not adequately control the Type I error rate in the presence of dependent effect sizes, which is the case in our dataset. By contrast, the two variants of Egger's regression (PET and PEESE) function effectively with RVE or MLMA according to the same authors. We thus applied PET-PEESE (variance-based, Stanley & Doucouliagos, 2014, and sample size-based, Pustejovsky & Rodgers, 2019), as suggested. We report full results in the Supplementary Table 15, and we summarize them in the main manuscript:

Publication bias was assessed using multilevel Egger regression^{50,51} and two variants of PET-PEESE^{52,53}. Detailed results are presented in Supplementary Tables 14-15, and funnel plots are presented in Supplementary Figure 1.

For each main category of predictors (see Results), effect sizes were regressed on their standard errors (Egger regression⁵¹, or PET model), and on either sample size⁵² or variance⁵³ (PEESE) in a three-level model. In the case of Egger regression, results suggested presence of publication bias in the case of the associations of cultural distance with psychological adaptation, and exposure to social groups with psychological adaptation, which might indicate that these effects are inflated. In the case of PET-PEESE, meta-regression intercepts indicate the size of unbiased effects. If the intercept of the PET model test is non-significant at $\alpha = .10$, PET results should be interpreted; if the intercept of the PET test is significant, PEESE results should be interpreted⁵³. In all analyses, the estimated unbiased effects were consistent with the results of the core analyses presented here, except for the variance-based PEESE⁵³ which yielded larger effects of all predictors for socio-cultural adaptation. Therefore, PET-PEESE analyses did not show evidence of noteworthy bias.

References:

- Rodgers, M. A., & Pustejovsky, J. E. (2021). Evaluating meta-analytic methods to detect selective reporting in the presence of dependent effect sizes. *Psychological Methods*, 26(2), 141.
- Pustejovsky, J. E., & Rodgers, M. A. (2019). Testing for funnel plot asymmetry of standardized mean differences. *Research Synthesis Methods*, 10(1), 57-71. <https://doi.org/10.1002/jrsm.1332>
- Stanley, T. D., & Doucouliagos, H. (2014). Meta-regression approximations to reduce publication selection bias. *Research Synthesis Methods*, 5(1), 60-78.

R3.4. *Additionally, p-hacking, which is distinct from publication bias (see work by Maya Mathur on this topic), should be addressed. Currently, two existing techniques can correct for both p-hacking and publication bias: RTMA and MAIVE. The authors should consider applying at least one of these methods. Given the potential for inflated effect sizes, correcting for these biases could significantly alter the study's conclusions.*

Response to R3.4. We thank the Reviewer for these additional suggestions. We have carefully reviewed both recommended methods, and we summarize our conclusions below.

(a) *Mathur's right-truncated meta-analysis (RTMA) method and phacking R package:* RTMA involves analyzing only the published non-affirmative results to essentially impute the full underlying distribution of all results prior to selection due to p -hacking and/or publication bias. Yet, this method works under the assumption of independent effect sizes, which is not valid for our data. We utilized a diagnostic quantile-quantile plot for the right-truncated meta-analysis (RTMA). However, since the points do not align closely along a 45-degree line, the RTMA method was unsuitable for our dataset. Please see below for the degree of divergence.

For socio-cultural adaptation:

```
## P hacking  
library(phacking)  
  
data_sc_rtma <- phacking_meta(data_sc$yi_a, data_sc$vi, parallelize = TRUE)  
rtma_qqplot(data_sc_rtma)
```

For psychological adaptation:

```
data_pa_rtma <- phacking_meta(data_pa$yi_a, data_pa$vi, parallelize = TRUE)  
rtma_qqplot(data_pa_rtma)
```

(b) *Irsova et al.'s MAIVE method and R package:*

We have considered implementing this method. We think that it shows promise, and it is definitely worth considering for future use. Unfortunately, at this point both the method and the R package are brand new and have not been thoroughly tested and validated yet. If we understand correctly, the paper is still under peer review, and the package, based on our tentative test, displays some inconsistencies. Specifically, the installation method indicated in the README (`devtools::install_github("meta-analysis-es/meta_maive")`) does not seem to work, we thus used a different method (`devtools::install_github("meta-analysis-es/maive")`) which seemed to point to the correct repository. Nevertheless, we found further challenges, namely the function arguments and document arguments were mismatched (Function arguments: ``dat``, ``method``, ``weight``, ``instrument``, ``studylevel``, and ``AR``; Document arguments: ``dat``, ``method``, ``weighting``, ``instrumenting``, ``AR``, and ``correlation``). We have attempted to run some analyses, but provided the above mismatch, we were uncertain whether we specified the models correctly. By consequence, we could not be sure of the accuracy of the results, and we refrained from reporting them here.

References:

Mathur, M. B. (2024). P-hacking in meta-analyses: A formalization and new meta-analytic methods. *Research Synthesis Methods*.

Irsova Z., P. Bom, T. Havranek, & H. Rächinger (2024): Spurious Precision in Meta-Analysis of Observational Research. *Nature Human Behaviour*, 2nd revision

R3.5. 2) Meta-analysis methods

The paper does not clearly specify which meta-analysis technique is used. Is it the standard random-effects model with inverse-variance weighting? Fixed effects? Are weights based on sample size or not applied at all?

This omission is critical because in meta-analyses of observational studies, reported precision can be misleading. If inverse-variance weights are used without adjusting for spurious precision, the estimates may be biased. The supplementary information on methodology requires significantly more detail, and key methodological choices should be outlined in the main text as well.

Response to R3.5. As requested, we now describe our analytic approach in more detail on p. 32:

To account for correlated effects (i.e., multiple included effects from one primary participant sample), all analyses were conducted using three-level meta-analytical model with inverse variance weights, where Level 1 corresponded with variation between participants, Level 2 with the variation of effect within each independent sample, and Level 3 with the variation of effects between independent participant samples^{46,47}.

As stated above, we acknowledge Irsova et al. (2023)'s interesting discussion of how *p*-hacking practices can introduce biases in both the precision and coefficient estimates of the original studies used in meta-analyses. However, as we argued above, their method and package, still under review, has not yet been thoroughly tested. We therefore followed the standard, validated practices for multi-level meta-analysis and used a three-level mixed model with inverse variance-based weights. As to other specifications, we followed the procedure outlined by Cheung (2014, ref. 46), as cited in the main manuscript.

R3.6. 3) Causality

The meta-analysis synthesizes correlations, not causal effects. The authors acknowledge this on p. 6, yet their language throughout the paper often implies causality. For instance:

Abstract: "Understanding what factors contribute to the successful and equitable inclusion of migrants is a matter of highest urgency. Our meta-analysis helps pinpoint such factors."

Page 5: "Important antecedents of adaptation," "Factors crucial to migrant adaptation."

The authors must be much more cautious with causal language.

Response to R3.5. We thank the Reviewer for noting these problematic statements. We have now reworded them as follows:

*Understanding what factors **are associated with** the successful and equitable inclusion of migrants is a matter of highest urgency.*

*Important **correlates** of adaptation*

*Factors **robustly related** to migrant adaptation.*

We also checked the remainder of the manuscript for causal language and made linguistic changes whenever needed.

R3.6. *If they wish to discuss causality (which I encourage), they should focus on a subset of quasi-experimental studies that explicitly identify or try to identify causal effects.*

Response to R3.6. Unfortunately, our dataset does not include any quasi-experimental studies, and there are only five experimental (intervention) studies, which is a limitation of the field discussed previously in several other publications. We therefore refrain from discussing causality.

R3.7. *Furthermore, the risk-of-bias analysis is far more useful for causal inference than for correlational research, and I would also like to see more details on this front.*

Response to R3.7. We fully agree with the Reviewer. However, as they correctly noted, we have no grounds for causal inference as almost all evidence is correlational. This said, please note that we have published an in-depth analysis of risk of bias in this dataset as a separate chapter that we reference below.

Reference:

Vu, D.H., Kunst, J.R., Tong, R., & Bierwiazzonek, K. (*in press*). Methodological challenges in cross-cultural adaptation research: Insights from a large-scale meta-analysis. In: L. A. Arasaratnam-Smith (Ed.). *Handbook of Communication and Culture*. Edward Elgar Publishing. https://doi.org/10.31234/osf.io/sk6mj_v1

R3.8. *4) Heterogeneity*

The authors do acknowledge heterogeneity, but its levels are extreme (I2 approaching 100% in some cases).

Response to R3.8. We thank the Reviewer for raising this issue. We did not comment much on the I^2 values because we do not consider it an adequate proxy for absolute heterogeneity in our dataset. The reason is that our dataset consists largely of large studies (including several with $N > 10,000$), and I^2 depends on the size of sampling variances (or sample sizes), thus the larger the sample size (i.e., the smaller the sampling error), the larger the ratio of true heterogeneity to sampling error, and the larger the I^2 values. Thus, I^2 is unlikely to be informative here, we therefore decided to focus on tau values and the resulting prediction intervals instead, and we centered our discussion on the effects for which prediction intervals exclude 0.

R3.9. *I find it problematic to pool migrants, refugees, business expatriates, and international students together. While the authors do report separate meta-analyses for these groups, if they wish to combine estimates further, a more logical grouping would be:*

Students and expatriates together

Migrants and refugees together

Pooling all groups into a single estimate is not meaningful given the vast differences in migration experiences (and wealth).

Response to R3.9. We thank the Reviewer for this comment. We used a typology of migrant groups that has been long used in migration psychology (see Berry, 1997, ref. 17 in the manuscript) and thereby aligns most closely with the field. Nevertheless, for full transparency, we have now added analyses with groups combined as the Reviewer suggested, and we report them in the supplemental information file, Supplementary Table 6.

In the main manuscript we decided to maintain the pooled results based on the following arguments:

- (a) The psychological processes postulated by cross-cultural adaptation theory – stress-and-coping and culture learning– are theorized to be universal across the migrant populations (e.g., Ward et al., 2001, ref. 11; Bierwiazzonek & Waldzus, 2016, ref. 13), and one of our goals was to test this assumption. Indeed, the overall lack of meaningful differences between groups in our analyses seems to support this notion. Of course, we agree with the Reviewer that the groups differ in their experiences and wealth, but from a psychological point of view, that only means that different groups encounter different forms or degrees of the factors meta-analyzed here, but once they encounter them, they should be similarly affected by them. As an example, an expat may be less likely to encounter blatant discrimination (e.g., being refused rental housing based on a foreign-sounding name) than a refugee, but if it happens to either of them, their well-being may suffer similarly.
- (b) The distinctions between groups, although they make theoretical sense, are blurry in research practice, and we have discussed this issue extensively in an earlier publication (Bierwiazzonek & Waldzus, 2017, ref. 13 in the manuscript). The applied criteria tend to be loose, and the same people might be categorized as international students in one study and as migrants in another; for example, especially in the US context, the term “international students” often means simply students born outside of the country, which does not distinguish them from (studying) migrants. Further, the same people might be categorized as migrants in one study, and as expatriates in another because, with the rise of self-initiated expatriation and digital nomadism, the distinction between expats and skilled migrants has become unclear. Research has increasingly defined expatriates as people for whom working abroad was the main motive to move – which is arguably the case for many migrants. Finally, studies on migrants rarely list having a refugee status as an inclusion criterion, potentially including many refugees; and vice-versa, studies on refugees seldom require a legal refugee status, thereby including people who otherwise could be categorized as migrants. Thus, in research practice group distinctions are unlikely to reflect the difference in experiences and/or wealth the Reviewer mentions.
- (c) In line with these notions, heterogeneity is not reduced when we split the data into migrant groups.

R3.10. 5) Migrant characteristics

The paper would be much stronger if it examined how migrant characteristics (e.g., legal vs. undocumented status, country of origin, socioeconomic background) influence adaptation. The authors likely have access to relevant data for at least some of these variables. Including such an analysis would greatly enhance the study's impact.

Response to R3.10. We thank the Reviewer for these suggestions. As suggested, legal vs. undocumented status, as well as socio-economic status have been included in our analyses (see Figure 4, variables: visa status, 1 – undocumented, 0 – documented; and low SES).

As to the country of origin, while we agree on its importance, we believe that it requires a different approach, namely, the inclusion of country as a fourth level in the multilevel meta-analytical model. Yet, with our data, this would mean a cross-classified model to account not only for the country participants come from, but also the country they reside in, as we would argue that the combination of both is what actually may impact adaptation outcomes. Such a model would require a very different approach to the data and data preparation. Moreover, it would call for applying country-level moderators to obtain meaningful insights. Since the results presented in the current paper are already dense and complex, we consider that such an analysis warrants a separate publication.

R3.11. 6) Style

The paper is competently written, but some wording could be simplified to improve accessibility. Additionally, the authors repeatedly emphasize that their research addresses a topic of high societal relevance. While true, these repeated statements could be trimmed for conciseness.

Response to R3.11. We thank the Reviewer for noticing this repetition. We have now removed most uniqueness and relevance statements, and ensured societal relevance is only mentioned once in the introduction. We also attempted to simplify some of the wording in the Results and Methods.

R3.11. The introduction to the Results section is difficult to follow. For example, the phrase "intercept-only models" will be unclear to many readers.

Response to R3.11. We have now edited the introduction to the Results section to make it more readable, and we have removed the phrase "intercept-only models." Instead, we refer to average effects (p. 6):

Whenever we refer to average effects obtained by subsetting the data, 95% confidence intervals around the mean effect are presented in the figures, indicating the precision of the estimate, and 90% prediction intervals, referring to the distribution of the population effect, are reported in the text. If both boundaries of a prediction interval are on the same side of zero, at least 90% of real-world effects can be expected to be so as well; if they are not, some real-world effects might go in opposing directions. Whenever we refer to comparisons between different categories of variables conducted as meta-regressions, only confidence intervals are reported because prediction intervals are not relevant for determining the significance of such comparisons. Importantly, our usage of the terms "effect" or "effect size" in the description of results is strictly due to meta-analytical conventions and is not intended to suggest causality (a point we return to in the discussion).

R3.12. *The results are presented monotonously, listing numbers for different categories without sufficient context or comparisons. Restructuring this section to highlight key contrasts would improve readability.*

Response to R3.12. We appreciate the Reviewer’s feedback on the presentation of our results. We understand that readers have diverging preferences when it comes to the tradeoff between comprehensive numerical reporting and narrative flow. Given the complex nature of our results (two distinct outcomes and multiple categories of measures), we opted for an approach that we ourselves as readers would find most effective for clear communication. Specifically, we chose to minimize linguistic diversity to ensure a streamlined presentation of the numerical findings, which we believe helps readers make direct comparisons across categories without unnecessary verbal variation obscuring the patterns. However, based on this feedback, we have improved transitions between sections to enhance readability while maintaining the systematic presentation of our findings.

R3.13. *7) Tables and figures*

All tables and figures should be self-explanatory, but some are not. For example, Table 1 is difficult to interpret. Readers struggle to understand what each row represents. More informative captions, notes, and clearer labeling would improve clarity.

Response to R3.13. We thank the Reviewer for this comment. We have now rewritten the captions, notes and labelling for Table 1 and all figures. For instance, we have now written out row and column names in Table 1 as below (see also pp. 7-8). Please note that most of those changes could not be tracked for technical reasons.

Table 1.

Characteristics of primary studies included in the meta-analysis. The table displays the number of studies, participant demographics, and mean length of stay in primary studies grouped by migrant group (migrants, expatriates, international students, refugees), by the type of investigated outcomes (socio-cultural and psychological adaptation), and by the available correlates of adaptation (cultural distance, exposure, social resources, stressors). Please note that one study might be counted multiple times (e.g., if it reports both outcomes and/or multiple correlates).

Studies on:	Studies with socio-cultural adaptation as the outcome				Studies with psychological adaptation as the outcome			
	Cultural distance	Exposure	Social resources	Stressors	Cultural distance	Exposure	Social resources	Stressors
All groups together								
Number of studies (k)	98	79	180	210	39	144	401	719
Number of participants (N)	20 900	17 290	44 535	52 149	16 001	148 722	223 687	470 588
% of male participants	60.13	51.93	57.27	53.47	54.38	46.09	46.39	46.02
Mean age of participants	34.63	30.15	33.18	31.16	31.41	35.48	32.76	33.56
Mean length of stay in the receiving country (months)	37.65	39.67	48.66	49.34	30.66	108.97	82.02	95.19
Number of countries of origin	17	14	27	36	7	28	47	62

R3.14. Despite these issues, I want to emphasize that this meta-analysis is fundamentally well-grounded: particularly in terms of data collection and literature search. A significant amount of effort has been invested in this paper, and I strongly believe that a revised version will be an extremely valuable contribution to both academia and policy discussions. With the necessary corrections, this study has the potential to become a highly cited, influential piece of research.

Response to R3.14. We are grateful for the Reviewer's constructive feedback and encouragement.

ITEMIZED REVIEWER COMMENTS

Reviewer 1

RI.1. The authors have done a very good job in addressing the comments raised by me earlier. I appreciate the limitations that the authors experienced in navigating the format of the journal (and taking a moment to also add this in the cover letter, very helpful to highlight). I appreciate the additional details (specifically the heatmap, Figure 1, is an excellent addition).

Response to RI.1. We thank the Reviewer for their positive feedback.

RI.2. I disagree with the assessment of the authors that a further discussion of heterogeneity as a result of disciplines may not be relevant for the (broader) readership, as this is an issue that may apply to numerous other fields too. The authors state that the methodology and measures are fairly similar, but I find that difficult to follow, given that even within the cultural field there are different standards of reporting (reliabilities, translation/adaptation, etc.). I had no access to the chapter referenced.

Response to RI.2. Indeed, the statement that the methodology is similar across the meta-analyzed study might have been too general, for which we apologize. We meant that in our assessment, the methodology was similar enough for us to consider it comparable and meta-analyzable together. This prevented us from including measures from ethnography or economics; by consequence, all included studies use some variation of psychological measures. We agree, of course, that even within psychology there are differences (e.g., between organizational studies on expats and social-psychological studies on migrants), and if we refrain from discussing those further, it is because we simply do not have much to add to the discussion published previously and based on an earlier version of our dataset (Bierwiazzonek & Waldzus, 2016, ref. 13 in the main manuscript).

As to field practices that might affect (or bias) heterogeneity statistics, including reliability and validation of measures, please note that these are captured by our risk of bias assessment (Supplementary Table 13, ROB3, ROB4, ROB7). These analyses suggest that these aspects do **not** meaningfully affect heterogeneity.

Moreover, a detailed discussion of field practices is included in the chapter by Vu et al. (in press) that we, once again, link below for the Reviewer's convenience. We apologize that the Reviewer could not access it earlier. To make it more convenient for other readers as well, we summarized the main points of this discussion in Supplementary Text 3, to which we now point on p. 27 and p. 30:

Additionally, methodological issues found in the meta-analyzed literature, including field-specific practices and conceptual overlap between variables, are summarized in Supplementary Text 3.

The supplementary text itself reads:

Reporting practices:

- **Failure to report essential demographic details (e.g., age, sex/gender, country of origin).** Of the included studies, 98.66% clearly report both the migrant generation and subgroup (e.g., migrants, international students, refugees, expats). However, 11.01% do not report the mean age of participants, 3.95% omit sex/gender information, 41.6% fail to include the mean length of stay, and 0.25% do not provide information about participants' countries of origin or destination. In other words, nearly half of the studies lack at least one essential demographic detail.
- **Omission of information on the reliability and validity of variables used in analyses.** 7.28% of the included studies were flagged for having biased predictor measures and 4.57% for having biased outcome measures (e.g., Cronbach's alpha lower than .70 or noted issues with reliability or validity). 65.65% of studies lacked sufficient information to assess the reliability and/or validity of predictor measures, and 32.17% of outcome measures (e.g., unreported Cronbach's alpha, unreported scale items for face-validity evaluation, the use of one-item scales, lack of clarity regarding scale direction).
- **Failure to report other basic statistical information (i.e., bivariate correlations):** 43.53% of studies
- **Scarcity of publicly available datasets, indicating a lack of open science practices.**

Measurement:

- **Conceptual overlap between measures:** Common method bias (i.e., semantic and conceptual overlap between measures that are theoretically adaptation outcomes, e.g., homesickness, culture shock; and measures that assess adaptation antecedents, e.g., acculturative stress scales, loneliness) can be found in at least 15% of the included studies. Additionally, acculturative stressor measures include other concepts that are used as predictors of adaptation (e.g., subscales for perceived discrimination, language barrier) that are often combined into one composite scale.

Research designs:

- **Lack of designs allowing for causal inference.** 88.82% of included studies are cross-sectional observational studies, 10.76% are longitudinal, and 0.42% are experimental.

Resulting biases:

We conducted risk of bias analyses including all issues reported above under Reporting Practices (see Supplementary Table 13, criteria ROB1, ROB2, ROB3, ROB7, ROB8) and the common method bias reported above under Measurement (ROB4). These analyses found no clear evidence of bias in terms of results, that is, the effect size and heterogeneity showed no meaningful change after biased studies were removed from analysis. However, additional analyses reported by Vu et al. (in press) suggested that confounding due to common method bias (ROB4), but not other biases, led to inflated effect sizes.

Reference:

Vu, D. H., Kunst, J. R., Tong, R., & Bierwiazzonek, K. (in-press). Methodological challenges in cross-cultural adaptation research: Insights from a large-scale meta-analysis. In L. Arasaratnam-Smith (Ed.), *Handbook of Communication and Culture*. Edward Elgar Publishing. https://doi.org/10.31234/osf.io/sk6mj_v1

R1.3. Looking forward to seeing the manuscript in print.

Response to R1.3. Once again, thank you for the positive evaluation.

R1.4. Minor:

I think it would be useful to rephrase “the Global North seemed overrepresented in our data” to “the Global North is overrepresented in our data” and “This geographical imbalance may limit the generalizability of our findings” to “This geographical imbalance limits the generalizability of our findings”.

Response to R1.4. We have applied the changes suggested by the Reviewer.

R1.5. The section on the underrepresentation of the Global South transitions to the call for multi-level analyses (which I agree with), but I think the link does not work out as well, as this is more an issue of the number of studies needed.

Response to R1.5. We agreed with the Reviewer, and we edited the relevant sentence, which now reads (p. 28):

Future research should address this limitation by focusing more on the underrepresented regions and directly examining the role of country-level factors through comprehensive multi-level analyses, which would provide more nuanced insights into potential contextual variations in acculturation processes.

Reviewer 2

R2.1. I appreciate the authors’ efforts in revising the manuscript. The revised version has improved in several respects.

Response to R2.1. We thank the Reviewer for the positive assessment.

R2.2. However, several issues remain that require further clarification and revision to enhance the clarity and rigor of the study:

The reported average effects of social resources appear to be misleading. In Supplementary Table 1, social resources do not show significant effects overall. This discrepancy may be due to the fact that significant effects were observed only for specific sources of support—namely, from

supervisors (p. 15). Please revise the relevant sections to reflect this nuance accurately and correct the potentially misleading interpretation.

Response to R2.2. We apologize for this misunderstanding. Supplementary Table 1 reports a meta-regression that compares the size of the average effects of the different categories of correlates (akin to a moderation analysis). A non-significant result for social resources does therefore *not* mean that resources have no significant effect; instead, it means that the average effect does not differ significantly in size from the average effect of stressors (baseline category). We have now attempted to improve the clarity of this analysis by specifying it in the caption of Supplementary Table 1, which now reads:

Meta-regressions comparing the absolute values of overall cumulative estimates from the four main categories of predictors. A significant result (i.e., $p(r) < .05$) means that the average effect for a given category differs significantly in size from the reference category; a non-significant result indicates a lack of significant difference.

The same wording was used in all the remaining Supplementary Tables that reported similar meta-regressions to prevent further misunderstandings: Supplementary Table 3, 4, 6 and 9.

R2.3. The term acculturative stressors is described as an umbrella term (p. 13), yet no clear definition or operationalization is provided in the Method section, as the authors suggest. This is problematic, especially because the constructs commonly considered acculturative stressors (e.g., language barriers, discrimination) appear to be analyzed separately in the current study. For example, the Acculturative Stress Index (Noh et al.) used in the analysis (ST 12) includes domains such as language barriers, social isolation, discrimination, and lack of occupational opportunities—all of which are treated as distinct variables in this study. This overlap raises concerns about conceptual clarity and potential redundancy. A clearer explanation of how items were selected and categorized is necessary, as the current reporting may mislead readers.

Response to R2.3. We thank the Reviewer for highlighting this issue, and we apologize for our lack of clarity in this regard. On p. 13, we now explain in more detail when the category Acculturative Stressors was used:

*(...) the strongest negative correlation with psychological adaptation was found for acculturative stressors (an umbrella term referring to composite measures combining various stressors specific to migration into one score, **used only in cases when scores for those specific stressors were not reported**; see Supplementary Table 12).*

In Supplementary Table 12, we provide a broader explanation in the column “Remarks”, which reads:

*Many acculturative stressor scales include subscales such as perceived discrimination or language barrier, which in our meta-analysis are analyzed separately. Thus, **the acculturative stressor subcategory was only used in cases when a study only reported a composite score involving all subscales of an acculturative stressor scale and did not provide separate scores for the subscales of our interest.** In cases when separate scores were provided (e.g., perceived*

discrimination, language barrier), we instead extracted those separate scores and categorized them accordingly. Note that categorizations resulting from this coding strategy were fully independent (i.e., an effect could be coded as pertaining to acculturative stressors or perceived discrimination, but never both), ensuring that all subcategories could be used in the same meta-regression.

Further, we completely agree with the Reviewer that this results in a conceptual overlap that is problematic and needs to be acknowledged. We provided a broad discussion of this issue in a separate chapter (please see reference below) which we have linked below for the Reviewer's convenience. Additionally, we now summarize its contents in the supplementary materials (Supplementary Text 3), to which we now point on p. 27 and p. 30:

Additionally, methodological issues found in the meta-analyzed literature, including field-specific practices and conceptual overlap between variables, are summarized in Supplementary Text 3.

In Supplementary Text 3, the relevant paragraph reads:

*Methodological issues detected in the meta-analyzed literature (...)
Conceptual overlap between measures:
Common method bias (i.e., semantic and conceptual overlap between measures that are theoretically adaptation outcomes, e.g., depression, culture shock; and measures that assess adaptation antecedents, e.g., acculturative stress scales, loneliness) can be found in 15% of the included studies. Additionally, acculturative stressor measures include other concepts that are used as predictors of adaptation (e.g., subscales for perceived discrimination, language barrier) that are often combined into one composite scale.*

Reference:

Vu, D. H., Kunst, J. R., Tong, R., & Bierwiazzonek, K. (*in-press*). Methodological challenges in cross-cultural adaptation research: Insights from a large-scale meta-analysis. In L. Arasaratnam-Smith (Ed.), *Handbook of Communication and Culture*. Edward Elgar Publishing. https://doi.org/10.31234/osf.io/sk6mj_v1

R2.4. To strengthen the practical implications of the findings, and to avoid conceptual overlap similar to the issue above, the manuscript would benefit from including example items related to support from authority figures. While such support is shown to be beneficial, it remains unclear what kinds of support are most effective. At minimum, the authors could discuss whether the items reflect relationship-oriented support (e.g., emotional encouragement) or task-oriented support (e.g., administrative assistance). Clarifying the nature of this support would enhance the real-world relevance of the findings for organizational settings.

Response to R2.4. We thank the Reviewer for this comment. As suggested, we have now provided sample items for support from authority figures, as well as all other measures, in Supplementary Table 12.

Unfortunately, although we tried to categorize measures of support from organizations and supervisors into socioemotional support and instrumental support, it was impossible for a staggering majority of primary studies using these constructs because they either used measures

mixing both kinds of support or failed to provide enough information. There were only three effects that we were able to confidently classify as socio-emotional support, while the remaining effects were classified as unspecified (see the public data at <https://osf.io/kfn5x>, column Sourcer – Organization, Supervisor, column Mode: Socio-emotional, Unspecified).

By consequence, even though we agree with the Reviewer that a discussion of the kind of authority support associated with best outcomes would be informative, we are not in the position to provide it. We can only conclude, based on Supplementary Table 11, that for other sources of support the distinction between instrumental and socioemotional support did not play a major role. We now acknowledge this issue on p. 28:

Assuming that this effect reflects a causal relation – which is yet to be demonstrated – it might be easily actionable for intervention purposes: If support from a supervisor makes a considerable difference to migrant adaptation, then training supervisors to provide it in the most efficient way might go a long way. Unfortunately, it remains unclear what kind of support supervisors would be best advised to focus on, as most instruments used to assess supervisory support combined socio-emotional and instrumental elements or lacked enough detail for clear classification (see Supplementary Table 12). Future studies that isolate these specific forms of support and test how each predicts migrant adaptation would greatly advance this under-researched area.

R2.5. The relatively weak or non-significant effects of both perceived and objective cultural distance warrant more careful interpretation. It is surprising that cultural distance does not emerge as an important factor in the experiences of migrants and refugees. Readers would benefit from a clearer description of how cultural distance was measured, including how the relevant questions were presented to participants. It would be particularly helpful to list the full set of items used in each scale. If this is not feasible, providing representative example items would still add clarity. While the authors briefly consider the possibility of subgroup variation (p. 24), further discussion on the potential measurement limitations and the need for more robust assessment would strengthen the manuscript's contribution.

Response to R2.5. We appreciate the Reviewer's comment. Please note, however, that the largest effect in this category (self-rated distance correlated with socio-cultural adaptation, $r = -.20$, $p < .001$; Figure 8; Supplementary Table 8), is consistent with the theoretical prediction that cultural distance matters most for a migrant's socio-cultural functioning within the new context, and less for a migrant's well-being. We have now added it to the discussion, p. 25:

*The role of variables theorized to be related to culture learning, namely exposure to social groups that can model or explain the new culture, and the cultural distance between the new culture and one's heritage culture, turned out to be more limited yet still having effects for both dimensions of adaptation. Of note, and in line with theoretical predictions, these effects were slightly larger for socio-cultural adaptation, **especially in the case of self-rated cultural distance and exposure to local nationals.***

We would also like to highlight that this size of effect is the norm in social psychological studies, according to new conventions, and some authors would consider it a medium-sized effect (see, for example, Lovakov & Agadullina, 2021).

Unfortunately, since our study is a meta-analysis, we are limited to the information from the primary studies, very few of which list the full set of items used to measure the constructs of interest, and even fewer specify how the relevant items were presented to participants. Thus, we are not in the position to fulfill the Reviewer's requests in this regard. However, to at least somewhat increase clarity, we have now added sample items of the eligible scale to Supplementary Table 12 as per the Reviewer's suggestion.

Reference:

Lovakov, A & Agadullina, ER (2021). Empirically derived guidelines for effect size interpretation in social psychology. *European Journal of Social Psychology*, 51, 485–504.
<https://doi.org/10.1002/ejsp.2752>

Reviewer 3

R3.1. The authors have successfully incorporated my suggestions, as well as those from the other referees. The revised manuscript is now much clearer, more transparent, and includes appropriate publication bias corrections using both PET-PEESE and a robust Bayesian model.

Response to R3.1. We thank the Reviewer for their positive feedback.

R3.2. My only remaining concern is the absence of any method that addresses p-hacking. This is particularly relevant in a meta-analysis of psychological studies, where selective reporting of significant results is a well-documented problem.

Methods such as RTMA or MAIVE are straightforward to implement and suitable for robustness checks, even in the presence of heterogeneity and dependent effect sizes. For example, working MAIVE code is available at https://meta-analysis.cz/maive/maive_R_package.zip. If they have any troubles running it, they can use any general IV package available in R or Stata. Of course, as the authors rightly point out, these methods rely on specific assumptions, but those can and should be explicitly discussed. Many papers have shown that p-hacking is important in observational research. Including at least one p-hacking robust estimator, even as a secondary analysis, would substantially strengthen the credibility of the findings and align the paper even more closely with current best practices in meta-research.

Response to R3.2. We thank the Reviewer for their concern. In response, we have now implemented the RTMA method. We report full results in Supplementary Table 16, and we summarize the results in the main manuscript, also discussing the bearing of assumption violations as suggested by the Reviewer (pp. 33-34):

Additionally, we tested for bias resulting from p-hacking by fitting right-truncated meta-analysis (RTMA)⁵⁵ models for all main categories of adaptation correlates (Supplementary Table 16). RTMA accounts for p-hacking by imputing, via Bayesian methods, the underlying distribution of population effects in order to estimate a corrected effect size. While RTMA is currently the only

fully validated p-hacking correction method, it assumes that effects are independent. Our data violate this assumption, thus the results should be taken with extreme caution. Indeed, in two cases (cultural distance for psychological adaptation, stressors for socio-cultural adaptation), anomalies in results suggested that RTMA estimates were invalid (see Supplementary Table 16 for details). For the remaining results, the comparison of the corrected effects to uncorrected ones suggested that our core analyses underestimated the true effect in two cases (social resources for psychological adaptation; cultural distance for socio-cultural adaptation, with $\Delta r = .15$ and $\Delta r = .35$, respectively), and overestimated them in four cases (exposure and stressors for psychological adaptation, exposure and resources for socio-cultural adaptation, with Δr ranging from .03 to .11).

Please note, however, that because of the assumption violations described in detail in the previous round of reviews, we refrained from reporting the corrected effects as our core results.

ITEMIZED REVIEWER COMMENTS

Reviewer 1

R1.1. I do not have further comments that require action and would like to thank the authors for the very nice job of incorporating the feedback I raised.

Response to R1.1. We thank the Reviewer for their positive feedback and comments throughout the revision process.

R1.2. Just in case it is helpful to the authors: I still cannot access the OSF link associated with the Vu et al *in press* paper (https://osf.io/sk6mj_v1?view_only=)

Page returns: "Page not found

The requested resource could not be found. If this should not have occurred and the issue persists, please report it to support@osf.io".

Response to R1.2. Indeed, the link is currently not working because OSF seems to have blocked this preprint without communicating the reason, and we have not been able to obtain their support with this issue to date. We apologize for the inconvenience. However, this chapter is now in press and has been cited as such in the main manuscript and the Supplementary Information. Because the full text should be available online very soon, we decided to take no further action to solve this issue.

Reviewer 2

R2.1. I have reviewed all the materials submitted after the revision. I find that my previous comments have been thoroughly addressed and well reflected in the revised documents. I am pleased to recommend the manuscript for acceptance and publication.

Response to R2.1. We thank the Reviewer for the positive assessment and constructive feedback throughout the revision process.

Reviewer 3

R3.1. The authors have improved the paper again and incorporated my additional comments. I believe the paper is now ready for publication.

Response to R3.1. We thank the Reviewer for their positive feedback and helpful comments throughout the revision process.